# A study on the drag coefficient in wave attenuation by vegetation

Zhilin Zhang[1,2,3,4,5], Bensheng Huang[1,2,3,4], Chao Tan[1,2,3,4], Xiangju Cheng[5],

[1] Guangdong Research Institute of Water Resources and Hydropower, Guangzhou, 510630, China
[2] State and Local Joint Engineering Laboratory of Estuarine and Hydraulic Technology, Guangzhou 510630, China
[3] Guangdong Provincial Science and Technology Collaborative Innovation Center for Water Safety, Guangzhou 510630, China
[4] Guangdong Key Laboratory of Hydrodynamic Research, Guangzhou 510630, China
[5] School of Civil Engineering and Transportation, South China University of Technology, Guangzhou, 510641, China

*Correspondence to*: Zhilin Zhang (zhilin_zhang@outlook.com)

**Abstract.** Vegetation in wetlands is a large-scale nature-based resource providing a myriad of services for human beings and
the environment, such as dissipating incoming wave energy and protecting coastal areas. For understanding wave height
attenuation by vegetation, there are two main traditional calibration approaches to the drag effect acting on the vegetation. One
of them is based on the rule that wave height decays through the vegetated area by a reciprocal function and another by an
exponential function. In both functions, the local wave height reduces with distance from the beginning of the vegetation
depending on a damping factor. These two damping factors which are usually obtained from calibration by measured local
wave height are linked to the drag coefficient and measurable parameters, respectively. So the drag coefficient that quantifies
the effect of the vegetation can be calculated by different methods, following by connecting this coefficient to hydraulic
parameters to make it predictable. In this study, two relations between these two damping factors and methods to calculate the
drag coefficient had been investigated by 99 laboratory experiments. Finally, relations between the drag coefficient and
relevant hydraulic parameters were analyzed. The results show that emergent conditions of the vegetation should be considered
when studying the drag coefficient; traditional methods which had overlooked this condition cannot perform well when the
vegetation was emerged. The new method based on the relation between these two damping factors performed as well as the
well-recognized method for emerged and submerged vegetation. Additionally, the Keulegan-Carpenter number can be a
suitable hydraulic parameter to predict the drag coefficient only the experimental setup especially the densities of the vegetation
can affect the prediction equations.

## 1 Introduction

To meet the current wave prevention requirements, it is practical to construct ecological safety barriers with wetland vegetation
based on natural conditions. Vegetation in wetlands can enhance the toughness of the coast and save construction investment
effectively by dissipating incoming wave energy (Reguero et al., 2018). Practice also has proved that vegetation in wetlands
can provide services such as enhancing coastal ecosystem and biodiversity, enhancing fisheries and forestry production,
increasing bank stability, and promoting tourism economy, whereas the vegetated area occupies land resources in floodplain

(Schaubroeck, 2017; Keesstra, 2018). Hence, it is necessary to better understand the mechanism of wave attenuation to promote the efficiency of the nature-based solution.

Wave attenuation by vegetation is mainly induced by the drag force provided by the vegetation acting on water motion, as investigated in different researches such as numerical modeling (e.g., Wu et al., 2016; Suzuki et al., 2019), laboratory experiment (e.g., Hu et al., 2014; Wu and Cox, 2015, 2016), or field study (e.g., Danielsen et al., 2005; Quartel et al., 2007). The drag force is closely related to the drag coefficient $C_D$ which quantifies the drag or resistance of vegetation in water (Chen et al., 2018). This coefficient is one of the most uncertain parameters in the complicated interaction between the vegetated area and water because the drag effect can be fairly different on various time and space scales.

The calibration method for the drag coefficient is based on the perspective of wave energy dissipation and wave height reduction which will be discussed in Section 2, while Dean (1979) and Kobayashi et al. (1993) proposed that local wave height decaying through the vegetated area following a reciprocal function and exponential function, respectively. These two calibration functions describe local wave height with a distance from the beginning of vegetation and a factor reflecting the damping, so the corresponding factor can be calibrated based on measured wave height through the vegetated area. The damping factor $\alpha'$ from the reciprocal function and the exponential damping factor $k'$ from the exponential function are often linked to the drag coefficient $C_D$ and measurable parameters such as water depth and density of stems. For instance, Dean (1979) proposed an equation to calculate $C_D$ based on the damping factor and the model had been developed by researchers such as Knutson et al. (1982), Dalrymple et al. (1984), and Losada et al. (2016). Overall, the drag coefficient can be calculated by calibrating $\alpha'$ or $k'$ using measured local wave height, then the researchers built non-linear relations between $C_D$ and hydraulic parameters such as the Reynolds number (e.g., Hu et al., 2014; He et al., 2019). In this way, the drag of vegetation in water becomes predictable based on the non-linear relations and the values of these hydraulic parameters under different operating conditions.

Zhang et al. (2021) had compared these two calibration approaches by these two featured functions directly and yielded a connection between $\alpha'$ and $k'$, then a new equation to calculate the drag coefficient had been revealed. However, Zhang et al. (2021) overlooked the relation between $k'$ and $C_D$ by Kobayashi et al. (1993) and only used the relation between $\alpha'$ and $C_D$ by Dean (1979). In this article, using the well documented relation between the damping factor $\alpha'$ and the drag coefficient $C_D$ by Dalrymple et al. (1984) as well as the mentioned relation by Kobayashi et al. (1993), these two traditional approaches had been compared from another perspective and the second connection between $\alpha'$ and $k'$ had been revealed.

Hence, there are two relations between the damping factor and the exponential damping factor from two perspectives, and they had been analyzed by 99 cases from collected data and experimental experiments in this study. Additionally, in normal tidal conditions and the initial stage of storm surge, vegetation in wetlands can be emerged while by storm surge, vegetation is

submerged or near-submerged. Existing methods to calculate the drag coefficient had been compared considering these emergence conditions. Finally, relations between $C_D$ and hydraulic parameters, for instance, the Reynolds number ($R_e$), the Keuglan-Carpenter number ($KC$), and the Ursell number ($Ur$), had been studied.

## 2 Theoretical foundations

Typically, the drag coefficient $C_D$ is determined from the perspective of wave energy dissipation, represented by the decay of
wave height. Dean (1979) proposed one of the first models for wave attenuation by vegetation in which wave height throughout the vegetated area can be expressed as a reciprocal function:

$$K_X = H(X)/H_0 = 1/(1 + \alpha'X), \tag{1}$$

where $K_X$ (-) is the relative wave height at a distance $X$ (m) through the vegetation field from the beginning of vegetation, $H(X)$ (m) is the local wave height, $H_0$ (m) is the incident wave height, and $\alpha'$ (m$^{-1}$) is the damping factor.

Based on empirical estimates of fluid drag forces acting on vertical, rigid cylinders, Dean (1979) found that:

$$\alpha' = C_D dNH_0/6\pi h, \tag{2}$$

where $d$ (m) is the diameter of the circular vegetation cylinder, $h$ (m) is the water depth, and $N$ (stems m$^{-2}$) is the average number of stems per unit area.

Then Dalrymple et al. (1984) formulated an algebraic dissipation equation practicing linear theory and conservation of wave energy where $\alpha'$ can be expressed as:

$$\alpha' = \frac{4}{9\pi} C_D N d_v k_w H_0 \frac{\sinh^3 k_w l_s + 3 \sinh k_w l_s}{\sinh k_w h (\sinh 2k_w h + 2k_w h)}, \tag{3}$$

where $d_v$ (m) is the vegetated area per unit height of plant normal to wave direction, $k_w$ (rad m$^{-1}$) is the wave number, and $l_s$
(m) is the submerged stem height.

On the other hand, Kobayashi et al. (1993) published that the local wave height decays exponentially through submerged artificial kelp:

$$K_X = H(X)/H_0 = \exp(-k'X), \tag{4}$$

where $k'$ (m$^{-1}$) is the exponential damping factor. Based on linear wave theory and the conservation equation of energy, $k'$ is expressed as (Kobayashi et al.,1993):

$$k' \cong \frac{1}{9\pi} C_D N d_v k_w H_0 \frac{\sinh 3k_w l_s + 9 \sinh k_w l_s}{\sinh k_w h (\sinh 2k_w h + 2k_w h)}. \tag{5}$$

If we compare these relations between the (exponential) damping factor and the drag coefficient (Eqs. (3) and (5)), a relation
between the damping factor $\alpha'$ and the exponential damping factor $k'$ can be derived:

$$\alpha'/k' \cong 1.$$ 
(6)

Recently, Zhang et al. (2021) presented a relation between $\alpha'$ and $k'$ looking at these featured functions (Eqs. (1) and (4)) directly. This method firstly scaled the distance $X$:

$$H/H_0 = 1/(1 + \alpha'X) = 1/(1 + \alpha x) = F(x),$$ 
(7)

and

$$H/H_0 = \exp(-k'X) = \exp(-kx) = G(x),$$ 
(8)

where $\alpha (= \alpha'L)$ (-) is the scaled damping factor, $L$ (m) is the length of vegetated area, $x (= X/L)$ (-) is the scaled distance through the vegetation field, $k (= k'L)$ (-) is the scaled exponential damping factor, and $F(x)$ and $G(x)$ represent functions.

Then by using the Taylor expansion, when the scaled distance $x$ equals half, the following equations had been derived:

$$F(x) = \frac{2}{\alpha+2} - \frac{4\alpha}{(\alpha+2)^2}(x - 1/2) + \frac{8\alpha^2}{(\alpha+2)^3}(x - 1/2)^2 - \frac{16\alpha^3}{(\alpha+2)^4}(x - 1/2)^3 + R_1(x),$$ 
(9)

and

$$G(x) = \frac{1}{e^{k/2}} - \frac{k}{e^{k/2}}(x - 1/2) + \frac{k^2}{2e^{k/2}}(x - 1/2)^2 - \frac{k^3}{6e^{k/2}}(x - 1/2)^3 + R_2(x),$$ 
(10)

where $R_1(x)$ and $R_2(x)$ are the residual terms. The relative magnitude of each term in Eqs. (9) and (10) had been analyzed by Zhang et al. (2021), and it had revealed that the first two terms on the right side of these equations are relatively large compared to other terms. Hence, considering only these two terms in Eqs. (9) and (10), the proportionality $\frac{2}{\alpha+2}/\frac{1}{e^{k/2}} = \frac{4\alpha}{(\alpha+2)^2}(x - 1/2)/\frac{k}{e^{k/2}}(x - 1/2)$ results in:

$$\alpha/k = 2/(2 - k),$$ 
(11)

which equals:

$$\alpha'/k' = 2/(2 - k'L).$$ 
(12)

Equations (6) and (12) have built bridges between the exponential function and reciprocal function, verifying that these two functions are reliable and capable to describe the wave height attenuation by vegetation satisfactorily. The rule of the attenuation is then limited by two functions, which can increase the reliability of the calibration.

However, application of Eq. (6) in Eq. (12) results in $k'L \cong 0$, which is not appropriate when there is vegetation in the wetlands. Hence, it is worth further studying the relation between these two damping factors to help us better understand the drag coefficient and wave attenuation by vegetation.

In addition, we had studied the relation between $C_D$ and three relevant hydraulic parameters, which are frequently used to model $C_D$, including: 1) the Reynolds number, $Re (= u_{max}d_v/\nu)$, where $\nu (=1.011 \times 10^{-6}$ m$^2$ s$^{-1})$ is the kinematic viscosity of

water and $u_{\max}$ (= $2\pi H_0/2T \tanh k_w h$) is the maximum horizontal wave velocity from linear wave theory, where $T$ (s) is the wave period; 2) the Keulegan-Carpenter number, $KC$ (= $u_{\max}T/d_v$), representing oscillatory flow around cylinders; and 3) the Ursell number, $Ur$ (= $\lambda^2 H_0/h^3$), characterizing the balance between wave steepness and the relative water depth, where $\lambda$ (m) is the wave length. Researchers had reported several formulas between $C_D$ and $Re$. For instance, Wu et al. (2011) obtained the following empirical equation:

$$C_d = 3.83 \times 10^{-6} + (5683/Re)^{1.17}. \tag{13}$$

Besides, He et al. (2019) revealed that

$$C_d = 18.025 \exp(-0.043KC). \tag{14}$$

Hence, the following two formulas are most possible solutions to study the non-linear relation between $C_D$ and these parameters:

$$C_D = a \exp(-b\bar{X}) \tag{15}$$

$$C_D = a + (b/\bar{X})^c \tag{16}$$

where $\bar{X}$ ~~could~~ can be $R_e$, $KC$ or $Ur$; $a, b, c$ are factors. Values of these factors can be obtained by the regression of $C_D$ by calibrated $\alpha'$ or $k'$ (Eq. (3) or (5)) and these parameters, and in this way, $C_D$ becomes predictable under different operation conditions. We had obtained the values of the factors and the corresponding adjusted R-square as in Section 5.4 by both equations, and it is hard to tell the difference between these results from Eqs. (~~13~~15) and (~~14~~16). The former ~~is~~ had been at last chosen because it contains less factors and is simpler than the latter.

## 3 Experimental setup and instrumentations

The experiments were conducted in a wave flume in Guangdong key laboratory of hydrodynamic research at Guangdong research institute of water resources and hydropower, China. The wave flume is 80.0 m long, 1.8 m wide, and 2.6 m deep (schematized in Fig. 1a, unit: m). The wave was generated by a wave generator at one end and absorbed at the opposite end.

The start of the vegetated area was located 52.7 m from the wave generator. The uniform vegetation was constructed by putting mimic plants (Fig. 1b) in holes drilled in the bottom. These two heights of mimic plants ($l_{vs}$) were 0.3 and 0.5 m and $d_v$ of the mimics was 0.057 m considering average diameters of the stem and leaves while the height ratio of them was about 0.5 (Fig. 1b). The three horizontal lengths of the vegetated area ($L$) were 4 m, 5 m, and 6 m, and two mimic stem densities ($N$) were 25 and 50 stems m$^{-2}$ (marked as N1 and N2, see Figs. 1c and 1d). The two water levels of the flume were 0.8 and 1.0 m so the corresponding water depths of the floodplain ($h$) were 0.3 m and 0.5 m.

The original wave height ($H_{\mathrm{ori}}$) of each designed regular wave was calibrated at 30 m from the wavemaker before these tests. In this study, seven wave gages (G1 to G7) were used to measure the wave height time series, which were placed 1 m apart

from each other from the beginning of the vegetated area (Fig. 1a) and the measurement at G1 was used as the incident wave

height ($H_0$) (Wu and Cox, 2015).

Control tests were carried out with no mimic plants to reduce the influence of flume bed and sidewalls. As listed in Table 1, sixteen ~~operating modes~~ cases were conducted including various conditions. Data of each test were collected during more than

200 s and each case was repeated for three times.

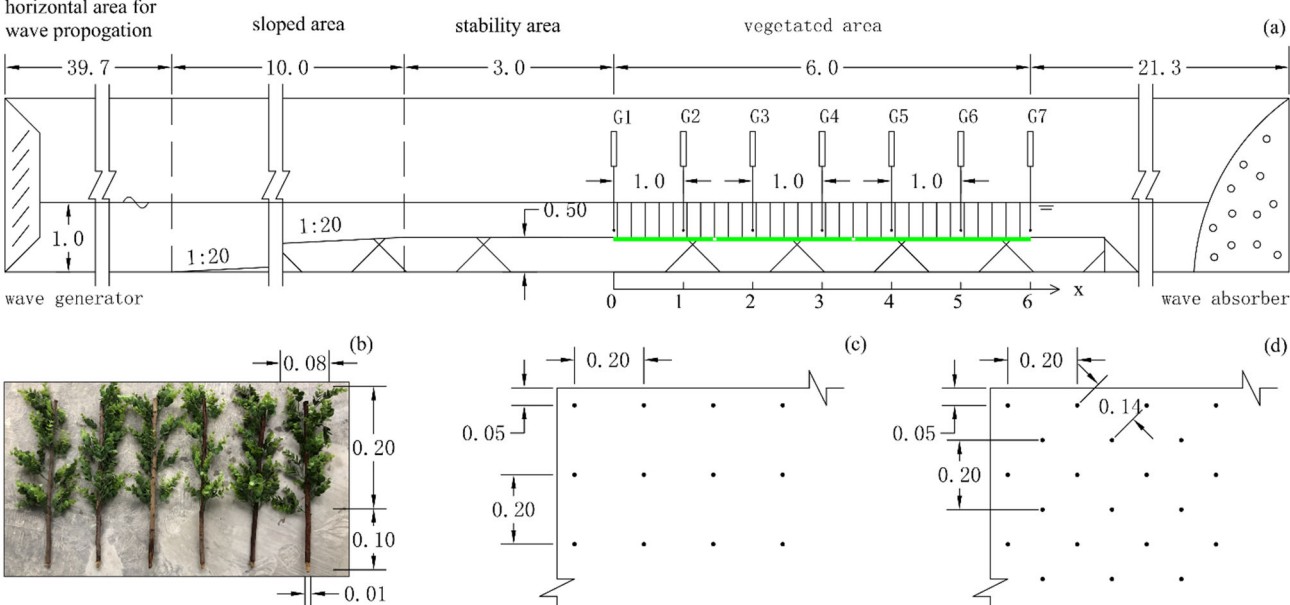

Figure 1: Experimental setup. (a) Schematic of the wave flume and instrument deployment, when the water depth of the floodplain was 0.5 m and mimic plants height was 0.5 m; (b) mimic plants with a height of 0.3 m; (c) and (d) top view of the mimic plant canopy with density of 25 and 50 stems m⁻².


Table 1: Hydrodynamic conditions with regular waves

| Cases | $h$ [m]/$H_{ori}$ [m] | $k_w$ [-] | wave period ($T$) [s] | $L$ [m] | $N$ [stems m⁻²] | $l_{vs}$ [m] |
|-------|------------------------|-----------|------------------------|---------|------------------|--------------|
| 1 | 0.3/0.12 | 2.24 | 1.00 | 4 | 25 | 0.3 |
| 2 | 0.3/0.12 | 2.24 | 1.00 | 5 | 25 | 0.3 |
| 3 | 0.3/0.12 | 2.24 | 1.00 | 6 | 25 | 0.3 |
| 4 | 0.3/0.12 | 2.24 | 1.00 | 4 | 25 | 0.5 |
| 5 | 0.3/0.12 | 2.24 | 1.00 | 5 | 25 | 0.5 |
| 6 | 0.3/0.12 | 2.24 | 1.00 | 6 | 25 | 0.5 |
| 7 | 0.3/0.12 | 2.24 | 1.00 | 4 | 50 | 0.5 |
| 8 | 0.3/0.12 | 2.24 | 1.00 | 5 | 50 | 0.5 |

| Cases | $h$ [m]/$H_{ori}$ [m] | $k_w$ [-] | wave period ($T$) [s] | $L$ [m] | $N$ [stems m$^{-2}$] | $l_{vs}$ [m] |
|---|---|---|---|---|---|---|
| 9 | 0.3/0.15 | 2.04 | 1.10 | 4 | 50 | 0.5 |
| 10 | 0.3/0.15 | 2.04 | 1.10 | 5 | 50 | 0.5 |
| 11 | 0.5/0.15 | 1.79 | 1.12 | 4 | 25 | 0.3 |
| 12 | 0.5/0.15 | 1.79 | 1.12 | 5 | 25 | 0.3 |
| 13 | 0.5/0.15 | 1.79 | 1.12 | 6 | 25 | 0.3 |
| 14 | 0.5/0.15 | 1.79 | 1.12 | 4 | 25 | 0.5 |
| 15 | 0.5/0.15 | 1.79 | 1.12 | 5 | 25 | 0.5 |
| 16 | 0.5/0.15 | 1.79 | 1.12 | 6 | 25 | 0.5 |

## 4 Data collection

Besides experiments in this study, observations in published literatures had been collected from Hu et al. (2014), Wu et al. (2011), and Wu and Cox (2015, 2016) as Zhang et al. (2021) presented. The summarized experimental setup is shown in Table

2. Overall, different laboratory experiments with different operation conditions had been conducted by the researches.

**Table 2: Experimental conditions from references**

| Reference | Type of plant | Plant height/$l_v$ [m] | Plant diameter/ $d_v$ [m] | Plant density/ $N$ [stem m$^{-2}$] | Incident wave height/$H_0$ [m] | Length of vegetation/$L$ [m] | Depth of water/ $h$ [m] |
|---|---|---|---|---|---|---|---|
| Hu et al. (2014) | Stiff wooden rods | 0.36 | 0.01 | 62/139/556 (VD1/VD2/VD3) | 0.032~0.202 | 6 | 0.25/0.5 |
| Wu et al. (2011) | Birch dowels | 0.48/0.63 | 0.009 4 | 350/623 | 0.083/0.084/ 0.085 | 3.66 | 0.5 |
| Wu and Cox (2015) | Plastic strips | 0.14 | 0.005 | 2 100 | 0.014~0.042 | 1.8 | 0.12 |
| Wu and Cox (2016) | Plastic strips | 0.14 | 0.005 | 1 618 | 0.015~0.034 | 0.9 | 0.12 |

## 5 Results and discussion

### 5.1 Reduction of wave height

Wave height along the vegetated area is a significant index for wave attenuation by vegetation. The calibrated reductions of

wave height by three equations demonstrating two examples (Cases 13 and 16) are shown in Fig. 2. It is clear that Eqs. (7) and

(8) were reliable relations between the scaled distance and the relative wave height. Additionally, with the calibrated $k$ value from Eq. (8), we calculated the value of $\alpha$ according to Eq. (11). Applying the calculated $\alpha$ in Eq. (7), the calculated relative wave height, which was named by Eq. (11) in Fig. 2, was appliable to fit the measurements, which suggested that Eq. (11) is valid. Results also show that the larger the value of the scaled damping factors, the stronger the wave attenuates.

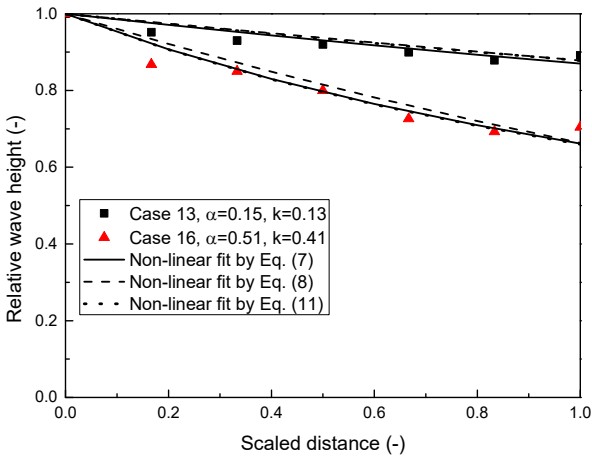


**Figure 2: Measured and predicted wave attenuation. Square and trigon symbols indicated measurements of Cases 13 and 16; solid, dashed and dotted lines represented the curves fitted by Eq. (7), Eq. (8), and Eq. (11).**

### 5.2. Relation between $\alpha$ and $k$

The relation between calibrated values of $\alpha$ and $k$ by 99 cases from this study and collected data is shown in Fig. 3. In the

study of Wu et al. (2011), Hu et al. (2014), and this research, both submerged and emerged cases had been conducted, and in the study of Wu and Cox (2015, 2016) the vegetation were emerged. The emerged and submerged cases had been separated for studying the influence of the emergent condition (emerged or submerged). Figure 3 showed shows that there is an obvious relation between $\alpha$ and $k$ for all cases. However, Eq. (6), which was obtained by comparing these relations between the (exponential) damping factor and the drag coefficient by Dalrymple et al. (1984) and Kobayashi et al. (1993), worked well

only when values of $\alpha$ and $k$ were smaller than around 0.4. Equation (1211), on the other hand, seemed a possible solution for the relation of these two factors, and the relation between $\alpha$ and $k$ is not strongly affected by the emergent condition even though these values are indeed relatively small when the vegetation is submerged ($0.04<\alpha<0.56$) than when it is emerged ($0.12<\alpha<1.43$). Notably, the analytical solution of Kobayashi et al. (1993), i.e., Eq. (5), was obtained and conducted using deeply submerged artificial kelp, and $H(X)^3 \cong H_0 H(X)^2$ was assumed which can only be valid when wave height reduces

slightly through submerged vegetated areas and the exponential damping factor is small. This is why Eq. (6) can only be profitable for submerged vegetation.

Equation (~~12~~11) also revealed that $\alpha - k = k^2/(2 - k) > 0$ since $k$ is smaller than 2 (Fig. 3). When the vegetation is deeply submerged, the calibrated $k$ close to zero and $\alpha$ is larger than but approximate to $k$ (Eq. (6)); when the vegetation becomes emerged, $\alpha$ and $k$ become relatively large and the difference between them enlarges, which can be seen in Figs. 2 and 3. That is to say, Fig. 3 shows that Eq. (~~12~~11) works well and it includes Eq. (6) to some extent.

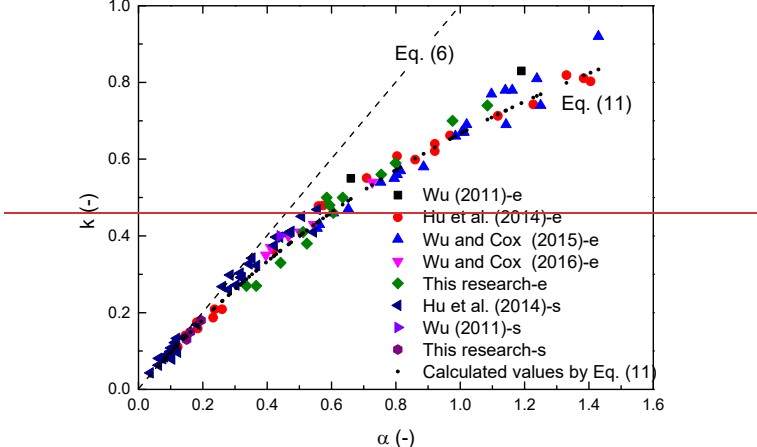

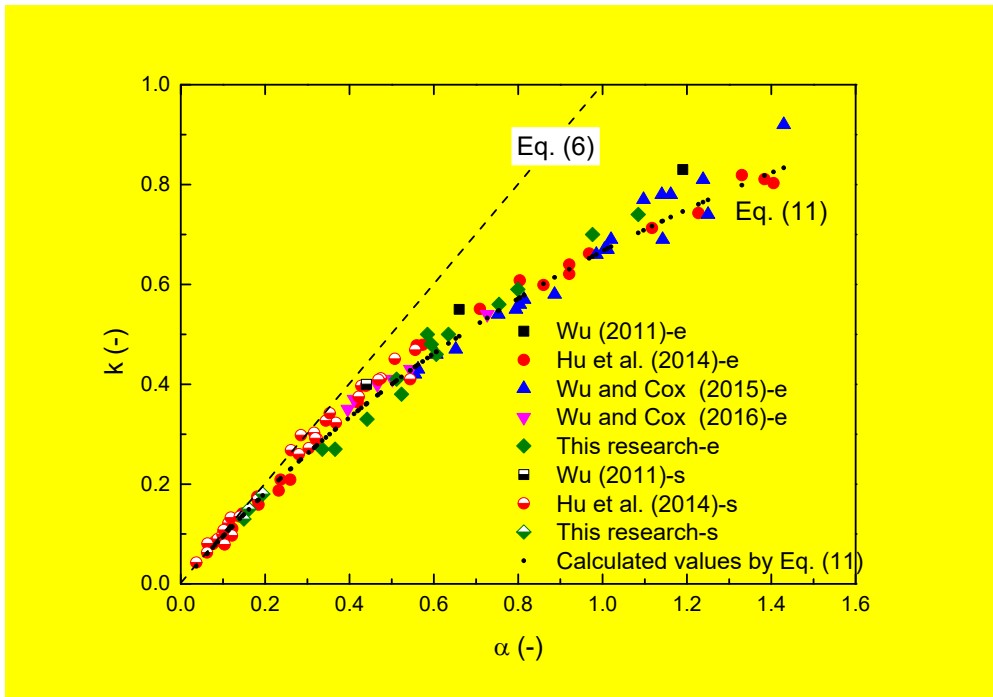

**Figure 3: Comparison of calibrated $\alpha$ and $k$. Different symbols indicated cases from different researches and emergent conditions. For emerged and submerged cases, "-e" and "-s" were are added after the references as the legend shown. The dashed and dotted lines indicated calculation by Eqs. (6) and (11), respectively.**

**5.3 Calculate $C_D$ by different methods**

**5.3.1 Calculate $C_D$ by Dean (1979)**

Several studies paid attention to the emergent condition of the vegetation recently. This condition (eg., by $l_s$) had been included in Eq. (3) by Dalrymple et al. (1984) while it had not been considered in Eq. (2) by Dean (1979). Both methods by Dean (1979) and Dalrymple et al. (1984) consider wave height decaying by the reciprocal function, in which the damping factor can be obtained by fitting the local wave height by Eq. (7). In this case, the value of the drag coefficient can be calculated using Eq. (2) or Eq. (3), and the comparison of results by these two equations is shown in Fig. 4. The result shows that these 99 cases obviously can be divided into two categories and they can be fitted by ==two linear lines==. Both the values of the adjusted R-square of the linear fit of emerged category and submerged category are 0.97, which means the results by these two equations are comparable. However, the slope of the former is about twice as large as the latter, so the emergent condition is necessary to be considered when calculating the drag coefficient in wave attenuation by vegetation. Additionally, the linear fit of the submerged category is close to the 1:1 line which means both equations are reliable and applicable for this category, while one of them is not suitable for emerged category considering the slope of the linear line. Since Eq. (3) had paid attention to the emergent condition, it is then regarded as a more satisfactory solution to calculate the drag coefficient for different conditions, while for emerged cases Eq. (2) can lead to larger values.

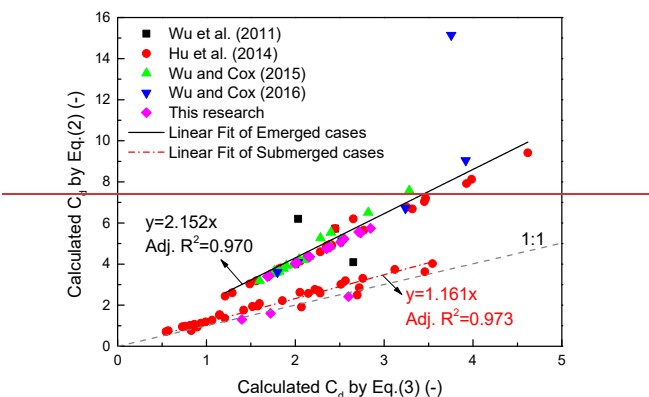

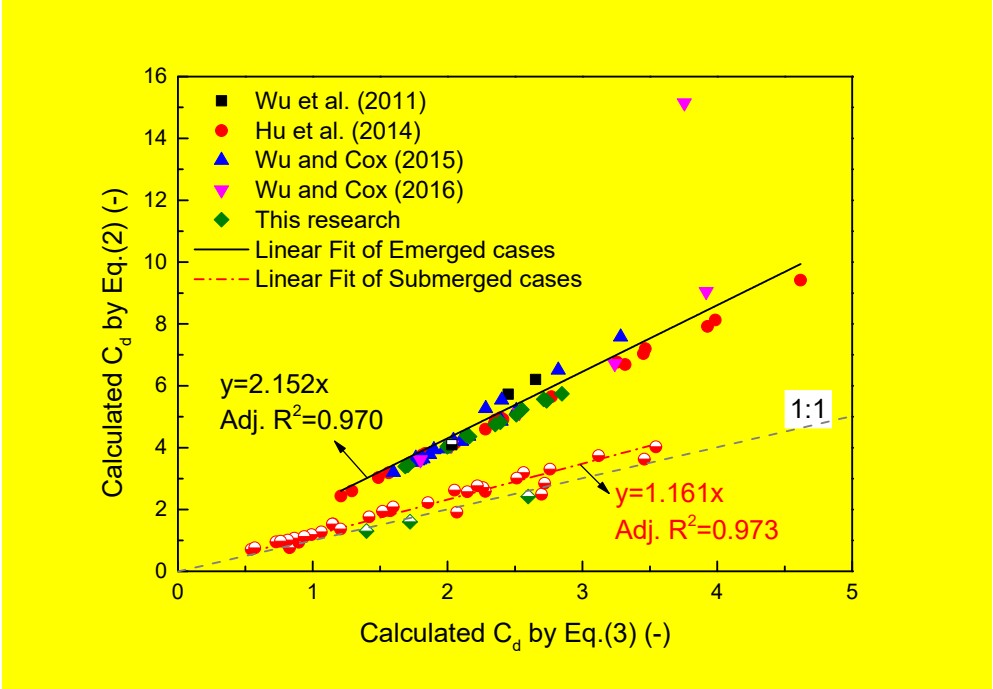

**Figure 4:** Comparison of the calculated values of $C_D$ by Eqs. (3) and (2). Different symbols ~~indicated~~ indicate cases from different researches, and partially and fully solid symbols denote submerged and emerged cases, respectively. The solid and dashed dot lines ~~indicated~~ indicate linear fit of emerged and submerged categories.

### 5.3.2 Calculate $C_D$ by Kobayashi et al. (1993)

Equation (5) by Kobayashi et al. (1993) also considered the emergent condition and it was obtained by using local wave height decaying exponentially. Hence, in this part, the comparison of the values of the drag coefficient by Eqs. (3) and (5) was studied to learn the influence of different decaying functions and the result is shown in Fig. 5. The value of $C_D$ by Kobayashi et al. (1993) was obtained by calculating $C_D$ using Eq. (5) on the base of the calibrated exponential damping factor by fitting the local wave height using Eq. (8). Figure 5 reveals that $C_D$ by Eq. (5) is always smaller than $C_D$ by Eq. (3). Also, cases can be divided into two categories. For submerged cases, the drag coefficient by Eq. (5) is close to but slightly smaller than that by Eq. (3), with a slope of 0.96 in Fig. 5; for emerged cased, the former is more smaller than the latter when the drag coefficient is larger. This is consistent to the conclusion in Section 5.2 since $C_D$ has positive correlation with $\alpha$ and $k$. In a word, for calculating the drag coefficient in wave attenuation by submerged vegetation, both Eqs. (3) and (5) can be the solution. However, for emerged cases, Eq. (5) can lead to ~~smaller~~ underestimated values of the calibrated $C_D$.

Additionally, although the regression of data should not be linear since $k/\alpha = (2 - k)/2 < 1$ which is not a constant, if we obtain $C_D$ by calibrating the exponential function for emerged cases, we have a rapid assessment that the value will be approximate 77% of the needed value. Moreover, the result reveals that $k'/\alpha' \approx 0.77$. Combining Eq. (12), $k'L = k$

approximates to 0.46, then $K_X \approx 0.63$ at the end of the vegetation according to Eqs. (4) and (8). It means that the reduction rate (=1-$K_X$) of the wave height for the emerged cases is about 37%. Furthermore, if we apply $k \approx 0.46$ in Eq. (~~12~~11), $\alpha$ is about ~~0.53~~0.60 then $K_X \approx 0.63$~~0.65~~ according to Eqs. (1) and (7). Values of $K_X$ which ~~were~~ are close by $\alpha$ and $k$ can be used to assess the wave attenuation by emerged vegetation very preliminary.

Of course, several parameters can affect the drag effect. In this case, certain cases should be considered separately instead of to use the result from a regression by all the cases with different operating conditions, then the slope of the comparison between the calculated $C_D$ by Eqs. (3) and (5) will be different so the calculated relative wave height will be different.

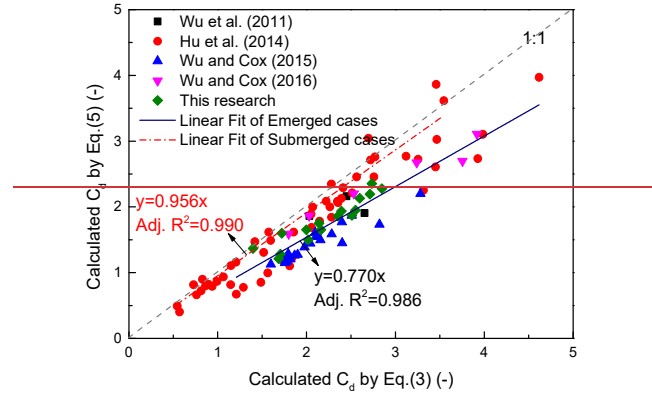

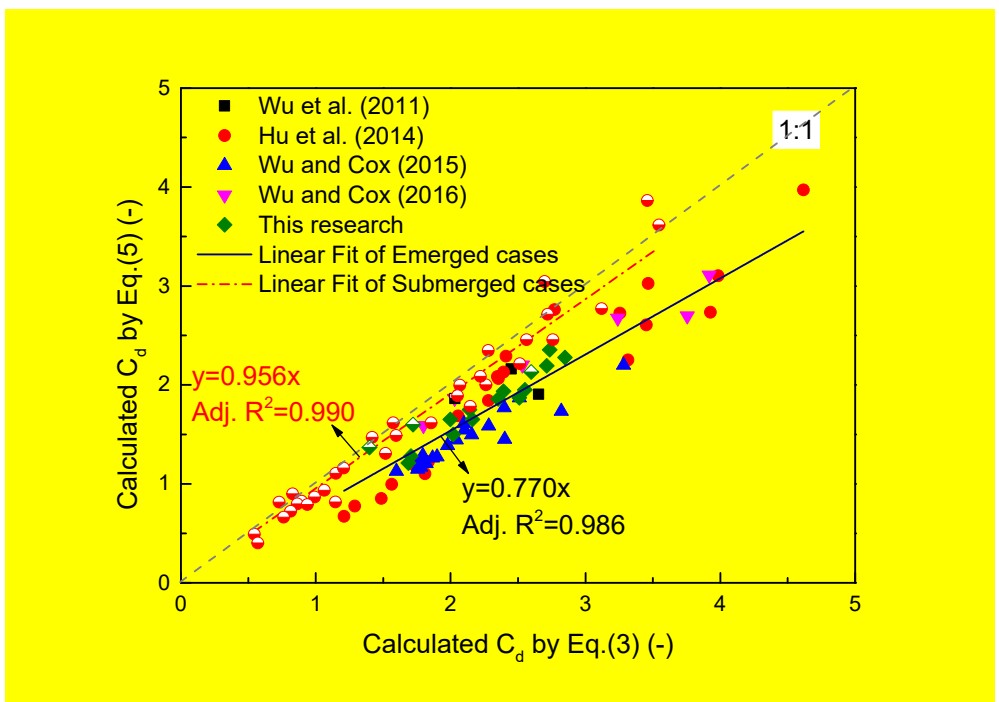

**Figure 5: Comparison of the calculated values of $C_D$ by Eqs. (3) and (5). Details are the same as Fig. 4.**

### 5.3.3 Calculate $C_D$ by a new method

The new method obtains the damping factor $\alpha'$ by using the calibrated $k'$ based on measured wave height and Eq. (12), so the drag coefficient $C_D$ can be calculated by Eq. (3). The Eq. (12)-based method used the rule that the local wave height decays exponentially and the classic relation between the damping factor and $C_D$ by Dalrymple et al. (1984). The comparison of the calculated values of $C_D$ by Eq. (3) and the new method is shown in Fig. 6. The result shows that there is a strong linear relationship among the calculated values in 99 cases from different researches. The slope of the linear fit is about unit and the adjusted R-square equals 0.99. The result is inspiring and shows the new method can lead to comparable results to the method by Dalrymple et al. (1984) for the drag coefficient. It is revealed that Eq. (12) is satisfactory and can be a bridge between the damping factor and the exponential damping factor. Based on the results in Figs. 5 and 6, the exponential damping factor $k'$ can be used to calculate $C_D$ while it needs to be converted to $\alpha'$ based on Eq. (12) instead of to be used directly in Eq. (5) for emerged cases; while for submerged cases, it can be a solution to calculate $C_D$ directly.

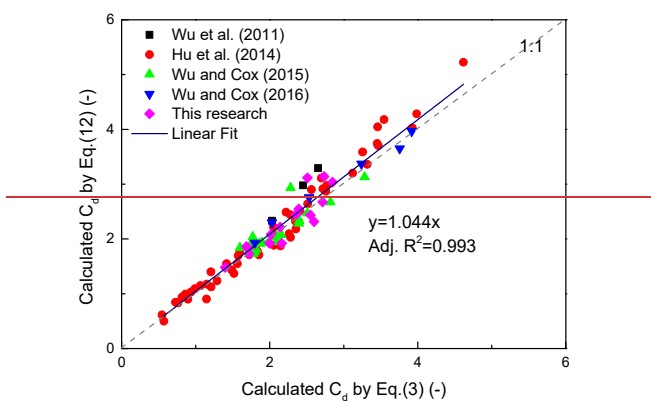

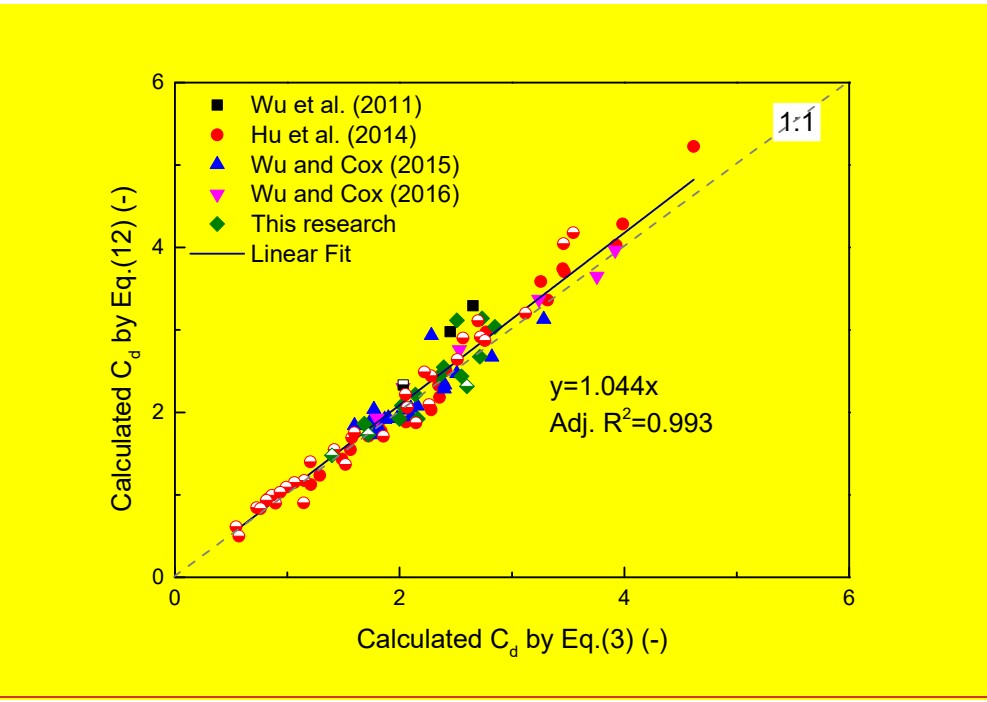

**Figure 6: Comparison of the calculated values of $C_D$ by Eq. (3) and the new method. Different symbols indicated cases from different researches. The solid line indicatesd linear fit of all cases.**

### 5.4. Relate $C_D$ to $R_e$, $KC$, and $Ur$

### 5.4.1. Relate $C_D$ to $R_e$

Relating the calculated $C_D$ by calibration method to $R_e$, $KC$, or $Ur$ is a common method to predict $C_D$. The relation between $R_e$ and the calibrated $C_D$ by the new method and the nonlinear fit by Eq. (15) are shown in Fig. 7. In the study by Hu et al. (2014) and this research, cases were grouped by different densities. The values of $R_e$ ranged from 370 to 38000 and the solid line following different groups of symbols can basically fit. Results reveals that separating cases from different densities is necessary for studying this relation while the effect of the emergent condition can be ignorable. Equation (15) was utilized to study this relation and the outcomes of the factors from nonlinear fit between $R_e$ and $C_D$ by the new method and by Eq. (3) are shown in Table 3. Results show that values for a certain factor ($a$ or $b$) based on the new method and Eq. (3) are close to each other especially for cases from Hu et al. (2014), supporting that the new method is comparable to Dalrymple et al. (1984). Moreover, values of factors can be quite different in various groups in Table 3 hence laboratory setup could play an important role on the relation between the drag coefficient and the Reynolds number. Hence, this relation is not universal for different cases. For example, the calculated line by Eq. (13) published by Wu et al. (2011) was is not very suitable for other groups of measurements. Hence, for engineering applications, case studies are needed for certain issues.

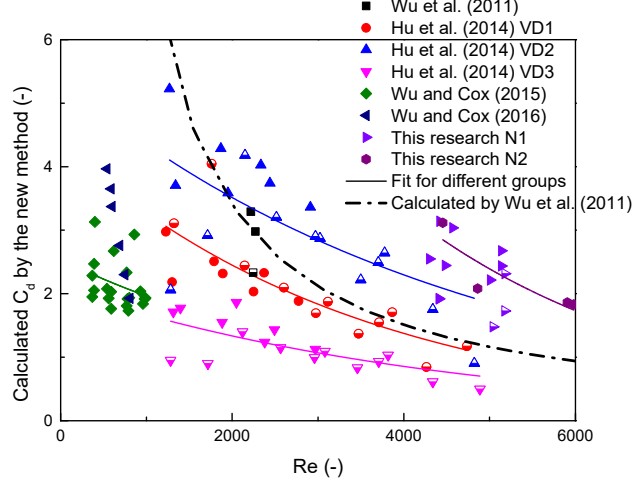

**Figure 7:** Relation between $R_e$ and $C_D$ by the new method. Different symbols indicate cases from different researches, and partially and fully solid symbols denote submerged and emerged cases, respectively. The solid lines following groups of the symbols indicate nonlinear fit by Eq. (15).

**Table 3: Outcome of the factors in Eq. (15) between $R_e$ and $C_D$ by the new method and by Eq. (3).**

| References | The new method | | | Equation (3) | | |
|---|---|---|---|---|---|---|
| | $a$ | $b$ | Adj. $R^2$ | $a$ | $b$ | Adj. $R^2$ |
| Hu et al. (2014) VD1 | 4.4 | $2.9 \times 10^{-4}$ | 0.65 | 4.0 | $2.5 \times 10^{-4}$ | 0.70 |
| Hu et al. (2014) VD2 | 5.4 | $2.1 \times 10^{-4}$ | 0.44 | 4.9 | $2.0 \times 10^{-4}$ | 0.45 |
| Hu et al. (2014) VD3 | 2.2 | $2.1 \times 10^{-4}$ | 0.47 | 2.1 | $2.4 \times 10^{-4}$ | 0.44 |
| Wu and Cox (2015) | 2.5 | $2.6 \times 10^{-4}$ | 0.04 | 3.0 | $5.3 \times 10^{-4}$ | 0.32 |
| This research N2 | 11.9 | $3.2 \times 10^{-4}$ | 0.65 | 7.2 | $2.5 \times 10^{-4}$ | 0.87 |

### 5.4.2. Relate $C_D$ to $KC$

The relation between $KC$ and $C_D$ by the new method is shown in Fig. 8. The values of $KC$ ranged from 9 to 130 and the range is much smaller than that of $R_e$ in Fig. 7. Similarly, Eqs. (1315) was utilized to study the relation between $KC$ and $C_D$ and outcomes of the factors are shown in Table 4. Results show that these fit lines are closer to each other than that in Fig. 7. The adjusted R-square values in Table 4 are overall larger than the corresponding numbers in Table 3. In addition, values for a certain factor based on these two methods were are closer to each other than the results in Table 3. From these studied cases, the Keulegan-Carpenter number can be a better parameter to describe the drag coefficient than $R_e$. Besides, for predicting $C_D$ by $KC$, factors in Eq. (15) can be different for different densities of vegetation and operation conditions, but the emergent

condition will not affect the result. ~~revealing that the new method perform well since the method by Dalrymple et al. (1984) is well-recognized.~~

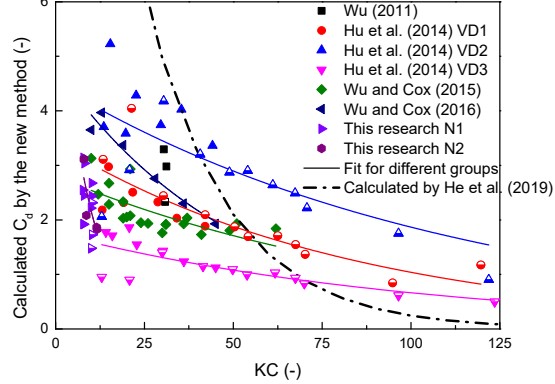

305

**Figure 8: Relation between $KC$ and the calculated $C_D$ by the new method. Details are the same as Fig. 7.**

**Table 4: Outcome of the factors in Eq. (15) between $KC$ and $C_D$ by the new method and ~~by~~ Eq. (3).**

| References | The new method | | | Equation (3) | | |
|---|---|---|---|---|---|---|
| | $a$ | $b$ | Adj. R$^2$ | $a$ | $b$ | Adj. R$^2$ |
| Hu et al. (2014) VD1 | 3.4 | $1.2 \times 10^{-2}$ | 0.66 | 3.2 | $1.0 \times 10^{-2}$ | 0.76 |
| Hu et al. (2014) VD2 | 4.5 | $8.8 \times 10^{-2}$ | 0.51 | 4.1 | $8.2 \times 10^{-3}$ | 0.52 |
| Hu et al. (2014) VD3 | 1.8 | $1.0 \times 10^{-2}$ | 0.58 | 1.8 | $1.0 \times 10^{-2}$ | 0.54 |
| Wu and Cox (2015) | 2.8 | $1.0 \times 10^{-2}$ | 0.44 | 3.1 | $1.5 \times 10^{-2}$ | 0.65 |
| Wu and Cox (2016) | 4.8 | $2.0 \times 10^{-2}$ | 0.94 | 5.0 | $2.4 \times 10^{-2}$ | 0.96 |
| This research N2 | 7.2 | $1.2 \times 10^{-1}$ | 0.54 | 5.0 | $9.4 \times 10^{-2}$ | 0.80 |

**5.4.3. Relate $C_D$ to $Ur$**

310    The relation between $C_D$ and the Ursell number $Ur$ had also been studied (Fig. 9). The values of $Ur$ ranged from 1 to 68. However, the nonlinear fit by Eqs. (15) ~~was~~ is unsatisfactory for all groups since the relation of these data is not so strong. Results show that comparing to $R_e$ and $KC$, $Ur$ is not a well-performed parameter for studying the drag coefficient in wave attenuation by vegetation.

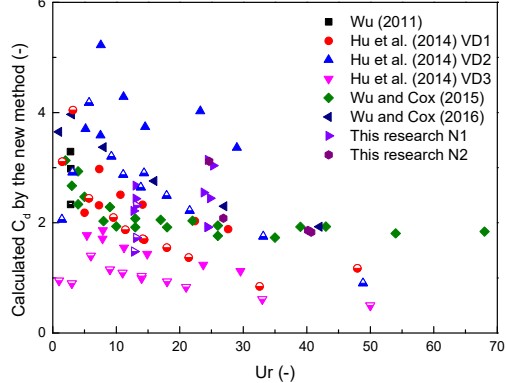

Figure 9: Relation between $Ur$ and the calculated $C_D$ by the new method. Details are the same as Fig. 7.

## 6. Discussion and conclusions

Wave attenuation by vegetation in wetlands is a large-scale nature-based solution providing a myriad of services for human beings. For understanding wave attenuation, two main traditional calibration approaches to the drag effect acting on the vegetation had been established, based on local wave height decaying by a reciprocal function or exponential function. These two reliable calibration methods by Dean (1979) and Kobayashi et al. (1993) can be combined from two perspectives: one by combining these featured functions directly (Eqs. (1) and (4)), and another by the relations between the (exponential) damping factor and the drag coefficient (Eqs. (3) and (5)). So, two relations between the damping factor $\alpha'$ and the exponential damping factor $k'$ had been derived (Eqs. (6) and (12)). Then, the relation between $\alpha'$ and $k'$ and the drag coefficient in wave attenuation were analyzed by 99 laboratory experiments. Furthermore, relations between $C_D$ and important hydraulic parameters ($Re$, $KC$, and $Ur$) were analyzed to make $C_D$ predictable under certain conditions.

The results showed that the reduction of wave height can be well described by both reciprocal and exponential functions. For submerged vegetation, which reduces wave height relatively slightly, the damping factor approximately equalled the exponential damping factor and Eq. (6) was applied. However, Eq. (12) was applicable no matter how submerged the vegetation was, which is a satisfactory result. Besides, for submerged vegetation, values of $C_D$ calculated by Eq. (2) by Dean (1979) and by Eq. (5) by Kobayashi et al. (1993) were consistent with the well-recognized Eq. (3) by Dalrymple et al. (1984). However, when the vegetation was emerged, Eqs. (2) and (5) were not in line with Eq. (3). On the other hand, the calcuated $C_D$ values by the new method by Zhang et al. (2021) in combination with Eq. (3) were almost the same as the results from the method of Dalrymple et al. (1984). Additionally, it is appeared that $KC$ performed best to predict $C_D$, better than $Re$ and $Ur$, although the -factors were different in different groups of laboratory observations. Therefore, further studies are needed in a variety of laboratory experiments.

Building a bridge between the two reliable methods by Dean (1979) and Kobayashi et al. (1993) is helpful. In this way, the reduction of wave height is limited by two functions so experimental outliers can be distinguished. Also, emergent conditions
and densities are very significant aspects to study the drag coefficient by vegetation. This method for the drag coefficient had been validated by a great amount of data under different laboratory conditions, however, the interaction between the vegetation and flow field is complicated and laboratory errors may affect the result so verification and/or calibration are needed further for predicting the drag coefficient.

**Data availability:** The data that support the findings of this study are available from the corresponding author upon reasonable request.

**Author contributions:** Z. Zhang, B. Huang, and C. Tan did the conceptualization and methodology. Z. Zhang and T. Chao did the data curation and formal analysis. Z. Zhang did validation and visualization. B. Huang and C. Tan did the funding
acquisition and project administration. B. Huang and X. Cheng did the supervision. All the authors contributed to writing and editing of the manuscript.

**Competing interests:** The authors declare that they have no conflict of interest.

### Acknowledgement

The authors especially thank Dr. Hu (Zhan) and Dr. Wu (Wei-cheng) for sharing laboratory data.
**Funding:** This work has supported by Guangzhou Science and Technology Program key projects [grant number 201806010143]; the Soft Science Research Program of Guangdong [grant number 2018B020207004]; the National Key Research and Development Program of China [grant number 2016YFC0402607].

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
