# Peer review of "A study on the drag coefficient in wave attenuation by vegetation"

_Hydrology and Earth System Sciences, 2021_

## Referee Comment (RC1)

*Review of **A study on the drag coefficient in wave attenuation by vegetation** by Zhang et al.*

The study of Zhang et al. focusses on wave attenuation by vegetation. Specifically, the authors look at an exponential and reciprocal function that describes wave height, and the accompanying damping factors. These damping factors both use the drag coefficient Cd, and the authors derive a new function that connects the two damping factors. Eventually, the authors predict Cd based on the different methods and damping factors. They conclude that the two damping factors are almost equal for submerged vegetation, but the new equation can be used for both submerged and emerged vegetation.

Generally, I like how the authors present their data, and find their conclusions well-supported with what they show. However, I still have several issues that the authors may want to improve on.

Something that confuses me throughout the whole manuscript, is that several steps to get to the drag coefficient are not clearly described. For example, in the results section (sect. 5.1 and 5.3.3) the authors say they calculated alpha, but then the value of k should be know. So how was this done exactly? See also my minor comments for more examples. Generally, I think it would be good if the authors add one extra section in the methodology, where the method related to each section in the results is explained in more detail. There, the authors could state specifically, and maybe even step-wise, which equation was used in which way.

I am also a bit confused about the Taylor expansions and how the authors arrive exactly at equations 11 and 12. This may be the lack of knowledge about this topic on my side, but I believe it is important to elaborate here and make it really clear to the reader what has been done and why.

I think the authors also need to elaborate the discussion. Especially the analyses to relate Cd to Re, KC and Ur are hardly discussed, and I think the authors should reflect here on the implications of their findings. Why is it important to link Cd to the parameters and what can we do with these findings? This is probably obvious for the authors, but it is good to also stress this for the reader. In addition, the authors should also make clear to the reader why the new equation is helpful and why the methods need to be linked. This is briefly done on page 16, but the main point seems here that you can use MS Excel, which is not a good argument in my view (there are so many free tools available, Python, R etc.). So please state clearly what new insights we gain from this and how it is helpful.

Lastly, my list of minor issues is still rather long and probably not even complete. Therefore, I would suggest that the authors go over the article again in full detail and try to improve their text.

To conclude, I like the study and believe the results are clear. Most of my comments are merely textual, so I believe the authors could easily improve the manuscript. I hope the authors find my comments useful, and I am looking forward to a revised version of the manuscript.

**Minor comments**

Abstract → The abstract should not refer to the main text, so it is better to remove the equation numbers.

P1.L16. Predicting → predict
P1.L24 of practical → of practical use?
P1.L24 barrier → barriers
P1.L29. What do you mean with floodplain resources?

P2.L32. Water motion in researches → water motion, as investigated in different researches
P2.L36. Complicate → complicated
P2.L39. Following → follow a
P2.L44. Later → that?
P2.L47. then→ that?
P3.L64. Vertical, rigid cylinder →  a vertical, rigid cylinder / vertical, rigid cylinders ?
P3.L66. Of circular → of the circular
P4.L100. What do you mean with the proportionality? I am not sure if I follow how you get to Eq. 11.
P4.L112. Understanding → understand
P4.L120. Why do you use that specific formula?
P4.L131. The three lengths of the canopies → here you mean the horizontal length, correct? So, x in Figure 1?
P4.L132. these → the
P4.L132. Depth → depths
P4.L140. List → listed
P5.L141. Collected more → collected during more?
P7.L151. Had shown → determined?
P7.L158. In laboratory → in a laboratory
P7.L173-174. Also...useful.→ How did you use Equation 11 here? How did you determine the k-value?
P8.L180. Was shown → is shown
P8.L187. did not strongly affect → was not strongly affected
P9.L198. Attention..recently.-→ Several studies paid attention to the emergent condition of the vegetation recently.
P9.L199. In this part...were compared. → How do you do this exactly? You fit equation 1 for alpha, and then calculate the drag coefficient Cd back with Eqs 2 and 3?
P9.L204. When study → when studying
P9.L213. Decaying function → decaying functions
P11.L223. How did you determine k here?
P11.L224. Decaying → decays
P12.L237. Different densities → what do you mean here?
P12.L238. Why are these considered as outliers?
P12.L238. Tendencies → what do you mean with tendencies here?
P12.L240. Due to → be due to
P12.L240. wave → waves
P12.L240. This might...were small. → This sounds like a bit of guessing, but you should be able to check this.
P12.L240. Results revealed...was ignorable. → How do I see this? Which density differences?
P12.L245. Various groups → do you mean the groups in Table 2? Please be specific.
P12.L247.  Case study is → case studies are?
P15.L275. By reciprocal → by a reciprocal
P15.L275. By combining...two perspectives. → These two reliable calibration methods by Dean (1979) and Kobayashi et al. (1993) can be combined from two perspectives:
P15.L277. These relations → the relations
P15.L300. Filed → field?

---

## Referee Comment (RC2)

**Review of "A study on the drag coefficient in wave attenuation by vegetation" by Zhang et al.**

In this study, the authors proposed a new hybrid method to link the two damping factors derived from two traditional approaches. Subsequently, the method was used to calibrate the drag coefficient and the relationship between the drag coefficient and relevant parameters (Re, KC, and Ur) were investigated. The paper is generally well-written. However, there are several major concerns that should be properly addressed before the paper can be considered to be accepted by this journal.

The major concerns:
1. The novelty of the manuscript: the authors mentioned that "Besides, based on local wave height, the exponential damping factor $k'$ can be obtained easily by MS Excel, while the damping factor $\alpha'$ needs professional numerical tools. Therefore, calculating $\alpha'$ by the calibrated $k'$ is much easier than calibrating $\alpha'$ directly by the well documented Eq. (3) which is the advantage of the new method in this study." I agree with the comments provided by the Reviewer#1 that this should not be the main novelty of this manuscript since the calibration of the damping factor $\alpha'$ is a standard procedure and can be easily conducted by commonly used software (such as Matlab or R language).
2. The methodology: It appears that the key Equations (7)-(12) in this manuscript have been derived in the previous study by the authors (Zhang et al., Acta Oceanol. Sin, 2021, in press). Thus, the main contribution lies in the study of the relation between the drag coefficient and three relevant hydraulic parameters? I would suggest the authors to clarify the relationship between their previous study and the current paper.
3. Figure 6: It appears that the proposed new method (Eq. 12) functions more or less the same as Eq. (3). Thus, with regard to the calibration of the dray coefficient, what's the difference between the new method and the method proposed by Dalrymple et al. (1984)?
4. The underlying mechanism and the difference between emerged and submerged conditions: one possible novelty could be the unified expression for the calibration of the drag coefficient both emerged and submerged conditions. However, can authors further explore the underlying mechanism and the difference between these two conditions by means of the new proposed method?

The minor comments:
1. Please carefully address all the minor comments provided by Reviewer#1.
2. Abstract: both equations and symbols should be avoided.
3. Figures 4-9: in both xlabel and ylabel, the Cd should be corrected as $C_D$
4. Section 4 data collection: Please reorganize this section, for the time being, the authors simply list the collected data.
5. Figure 3: in the legend, "Calculted"→ "Calculated"

---

## Author Comment (AC1)

We would like to thank referee #1 for the discussion.

The study of Zhang et al. focuses on wave attenuation by vegetation. Specifically, the authors look at an exponential and reciprocal function that describes wave height, and the accompanying damping factors. These damping factors both use the drag coefficient Cd, and the authors derive a new function that connects the two damping factors. Eventually, the authors predict Cd based on the different methods and damping factors. They conclude that the two damping factors are almost equal for submerged vegetation, but the new equation can be used for both submerged and emerged vegetation.
Generally, I like how the authors present their data, and find their conclusions well-supported with what they show. However, I still have several issues that the authors may want to improve on.

**1 Something that confuses me throughout the whole manuscript, is that several steps to get to the drag coefficient are not clearly described. For example, in the results section (sect. 5.1 and 5.3.3) the authors say they calculated alpha, but then the value of k should be known. So how was this done exactly? See also my minor comments for more examples. Generally, I think it would be good if the authors add one extra section in the methodology, where the method related to each section in the results is explained in more detail. There, the authors could state specifically, and maybe even step-wise, which equation was used in which way.**

■ Sentences (red marked) has been added or modified in the manuscript to show this methodology:

(1)P2,L41. Section 1. "These two calibration functions describe local wave height with a distance from the beginning of vegetation and a factor reflecting the damping, so the corresponding factor can be calibrated based on measured wave height through the vegetated area."

(2)P2,L45. Section 1. "Overall, the drag coefficient can be calculated by calibrating $\alpha'$ or $k'$ using measured data, then the researchers build non-linear relations between $C_D$ and hydraulic parameters such as the Reynolds number (e.g., Hu et al., 2014; He et al., 2019). In this way, the drag of vegetation in water becomes predictable based on the values of these hydraulic parameters under different operating conditions."

(3)P7,L173. Section 5.1. "Additionally, with the calibrated k value from Eq. (8), we can calculate the value of α according to Eq. (11). Applying the calculated α in Eq. (7), the calculated relative wave height, which is named by Eq. (11) in Fig. 2, is appliable to fit the measurement as the figure shown."

(4)P9,L199. Section 5.3.1. "Both methods by Dean (1979) and Dalrymple et al. (1984) consider wave height decaying by the reciprocal function, in which the damping factor can be calibrated by local wave height. In this case, the value of the drag coefficient can be

calculated using Eq. (2) or Eq. (3), and the comparison of results by these two equations is shown in Fig. 4."

(5)P10,L213. Section 5.3.2. "The calibrated value of $C_D$ by Kobayashi et al. (1993) was obtained by calculating $C_D$ using Eq. (5) on the base of the calibrated exponential damping factor. Figure 5 revealed that $C_D$ by Eq. (5) is always smaller than $C_D$ by Eq. (3)."

(6)P11,L223. Section 5.3.3. "The new method obtains the damping factor $\alpha'$ by using the calibrated $k'$ based on measured wave height and Eq. (12), so the drag coefficient $C_D$ can be calculated by Eq. (3)."

**2 I am also a bit confused about the Taylor expansions and how the authors arrive exactly at equations 11 and 12. This may be the lack of knowledge about this topic on my side, but I believe it is important to elaborate here and make it really clear to the reader what has been done and why.**

■ According the literature: the Taylor expansion is the standard technique used to obtain a linear or a quadratic approximation of a function of one variable. Recall that the Taylor expansion of a continuous function $f(x)$ is:

$$f(x) = f(a) + (x-a)f'(a) + (x-a)^2 \frac{f''(a)}{2!} + \cdots (x-a)^n \frac{f^{|n|}(a)}{n!} + \cdots R(x)$$

where $R(x)$ represents all the terms of higher order than a level, and a is a 'convenient' value at which to evaluate $f(x)$.

■ We scaled the reduction functions:

$$H/H_0 = 1/(1 + \alpha' X) = 1/(1 + \alpha x) = F(x), \tag{7}$$
$$H/H_0 = \exp(-k' X) = \exp(-kx) = G(x), \tag{8}$$

■ So by using Taylor expansion, Eqs. (7) and (8) can be derived as following when $x$ equals half:

$$F(x) = \frac{2}{\alpha+2} - \frac{4\alpha}{(\alpha+2)^2}(x-1/2) + \frac{8\alpha^2}{(\alpha+2)^3}(x-1/2)^2 - \frac{16\alpha^3}{(\alpha+2)^4}(x-1/2)^3 + R_1(x) \tag{9}$$

$$G(x) = \frac{1}{e^{k/2}} - \frac{k}{e^{k/2}}(x-1/2) + \frac{k^2}{2e^{k/2}}(x-1/2)^2 - \frac{k^3}{6e^{k/2}}(x-1/2)^3 + R_2(x), \tag{10}$$

■ Zhang et al. (2021) found that the first two terms of Eqs. (9) and (10) played the most significant role. Hence, considering only these two terms in Eqs. (9) and (10), the proportionality between the two first terms:

$$\frac{2}{\alpha+2} \Big/ \frac{1}{e^{k/2}} = \frac{4\alpha}{(\alpha+2)^2}(x-1/2) \Big/ \frac{k}{e^{k/2}}(x-1/2)$$

■ Formula simplification results in:

$$\alpha/k = 2/(2 - k), \tag{11}$$
which equals:
$$\alpha'/k' = 2/(2 - k'L), \tag{12}$$
Because $\alpha$ $(= \alpha'L)$ (-) is the scaled damping factor $\alpha'$, $k$ $(= k'L)$ (-) is the scaled exponential damping factor $k'$.

**3 I think the authors also need to elaborate the discussion. Especially the analyses to relate Cd to Re, KC and Ur are hardly discussed, and I think the authors should reflect here on the implications of their findings. Why is it important to link Cd to the parameters and what can we do with these findings? This is probably obvious for the authors, but it is good to also stress this for the reader.**

- Sentences has been added or modified in the manuscript to show this answer this question:

- (1)P2,L45. Section 1. "Overall, the drag coefficient can be calculated by calibrating $\alpha'$ or $k'$ using measured data, then the researchers build non-liner relations between $C_D$ and hydraulic parameters such as the Reynolds number (e.g., Hu et al., 2014; He et al., 2019). In this way, the drag of vegetation in water becomes predictable based on the values of these hydraulic parameters under different operating conditions."

- (2)The beginning of Section 5.4.1 in Page 12: "Relating calibrated $C_D$ to $R_e$, $KC$, or $Ur$ is a common method to predict $C_D$ without measured wave height."

**4 In addition, the authors should also make clear to the reader why the new equation is helpful and why the methods need to be linked. This is briefly done on page 16, but the main point seems here that you can use MS Excel, which is not a good argument in my view (there are so many free tools available, Python, R etc.). So please state clearly what new insights we gain from this and how it is helpful.**

- The result based on Eq. (12) is comparable to the classic Eq. (3), this is because the effectiveness of Eq. (12) which we want to study further in this manuscript. Previously, Cd can be calibrated by Eq. (3) based on the reciprocal function, and by Eq. (5) and the exponential function. Now we found the latter method is only applicable in submerged cases. But we can also use the exponential function to describe the decay of the wave height for emerged cases and calculate the drag coefficient by transforming $k$ to alpha by Eq. (12).

- The following paragraph has been added in Section 5.3.2, Page 10:

"For submerged cases, the drag coefficient by Eq. (5) is close to but smaller than that by Eq. (3), with a slope of 0.96 in Fig. 5; for emerged cased, the former is more smaller than the latter when the drag coefficient is larger. This is consistent to the conclusion in Section 5.2 since $C_D$ has positive correlation with $\alpha$ and $k$."

■ The following paragraph has been added at the end of Section 5.3.3, Page 11:

"Based on the results in Figs. 5 and 6, the exponential damping factor $k'$ can be used to calculate $C_D$ while it needs to be converted to $\alpha'$ based on Eq. (12) instead of to use $k'$ directly."

■ Please see the reply to Reviewer#2 for more detail.

**5 Lastly, my list of minor issues is still rather long and probably not even complete. Therefore, I would suggest that the authors go over the article again in full detail and try to improve their text.**
■ Thanks for the correction of language. The language has been checked and will be checked further.

To conclude, I like the study and believe the results are clear. Most of my comments are merely textual, so I believe the authors could easily improve the manuscript. I hope the authors find my comments useful, and I am looking forward to a revised version of the manuscript.

**Minor comments**
Abstract → The abstract should not refer to the main text, so it is better to remove the equation
numbers.
**1 P1.L16. Predicting → predict**
■ It has been edited.

**2 P1.L24 of practical → of practical use?**
■ It has been modified to "it is practical to …".

**3 P1.L24 barrier→ barriers**
■ It has been edited.

**4 P1.L29. What do you mean with floodplain resources?**
■ It has been modified to "land resources in floodplain".

**5 P2.L32. Water motion in researches → water motion, as investigated in different researches**
■ It has been edited.

**6 P2.L36. Complicate → complicated    P2.L39. Following → follow a**
■ It has been edited.

**7 P2.L44. Later → that?**
- It has been deleted.

**8 P2.L47. then→ that?**
- No modification required.

**9 P3.L64. Vertical, rigid cylinder → a vertical, rigid cylinder / vertical, rigid cylinders ?**
- It has been modified to "vertical, rigid cylinders".

**10 P3.L66. Of circular → of the circular**
- It has been edited.

**11 P4.L100. What do you mean with the proportionality? I am not sure if I follow how you get to Eq. 11.**
- This sentence has been edited:

"the proportionality between the two first terms $\frac{2}{\alpha+2} / \frac{1}{e^{k/2}} =$

$\frac{4\alpha}{(\alpha+2)^2}(x - 1/2) / \frac{k}{e^{k/2}}(x - 1/2)$ results in:"

**12 P4.L112. Understanding → understand**
- It has been edited.

**13 P4.L120. Why do you use that specific formula?**
- The last part of Section 2 has been edited:

"The following two formulas are most possible solutions to study the relation between $C_D$ and these parameters (He et al., 2019; Hu et al., 2014):

$C_d = a \exp(-b\bar{X})$ (13)

$C_d = a + (b/\bar{X})^c$ (14)

where $\bar{X}$ could be $R_e$, $KC$ or $Ur$; $a$, b, c are factors. Suitable values of these factors can be obtained by calibrated values of $C_d$, and in this way, $C_d$ becomes predictable by these parameters. We obtained the values of the factors and the corresponding adjusted R-square, and it is hard to tell the difference between the results from Eqs. (13) and (14) so the former is used because it is exponential and it has only two variable factors.

**14 P4.L131. The three lengths of the canopies → here you mean the horizontal length, correct? So, x in Figure 1?**
- Yes. "The three lengths of the canopies" has been modified to "The three horizontal lengths of the canopies".

**15 P4.L132. these → the**

■    It has been edited.

**16 P4.L132. Depth → depths**
■    It has been edited.

**17 P4.L140. List → listed**
■    It has been edited.

**18 P5.L141. Collected more → collected during more?**
■    Yes. It has been edited.

**19 P7.L151. Had shown → determined?**
■    It has been edited.

"Besides experiments in this study, observations in published literatures have been collected from Hu et al. (2014), Wu et al. (2011), and Wu and Cox (2015, 2016) as Zhang et al. (2021) presented."

**20 P7.L158. In laboratory → in a laboratory**
■    It has been edited.

**21 P7.L173-174. Also...useful.→ How did you use Equation 11 here? How did you determine the k-value?**
■    It has been edited.

**22 P8.L180. Was shown → is shown**
■    It has been edited.

**23 P8.L187. did not strongly affect → was not strongly affected**
■    It has been modified to "is not strongly affected".

**24 P9.L198. Attention..recently.-→ Several studies paid attention to the emergent condition of the vegetation recently.**
■    It has been edited.

**25 P9.L199. In this part...were compared. → How do you do this exactly? You fit equation 1 for alpha, and then calculate the drag coefficient Cd back with Eqs 2 and 3?**
■    Yes. The typical process to get the drag coefficient by the calibration method is like this: measure the local wave height H(X) through the vegetated area then calibrate the value of alpha or $k$ by Eqs. (7) or (8). Then Cd can be calculated by equations such as Eqs. (2), (3), (5). This is the calibrated value of Cd. Then the authors often relate the calibrated Cd to Re or KC, so the Cd is predictable when lack of measurement data. It has been modified:

"Both methods by Dean (1979) and Dalrymple et al. (1984) consider wave height decaying by the reciprocal function, in which the damping factor can be calibrated by local wave height. In this case, the value of the drag coefficient can be calculated using Eq. (2) or Eq. (3), and the comparison of results by these two equations is shown in Fig. 4."

**26 P9.L204. When study → when studying**
■ It has been edited.

**27 P9.L213. Decaying function → decaying functions**
■ It has been edited.

**28 P11.L223. How did you determine k here?**
■ k is obtained by calibrating the measured wave height through the vegetation by Eq. (8). This sentence has been modified:

"The new method obtains the damping factor $\alpha'$ by using the calibrated $k'$ based on measured wave height and Eq. (12), so the drag coefficient $C_D$ can be calculated by Eq. (3)."

**29 P11.L224. Decaying → decays**
■ It has been edited.

**30 P12.L237. Different densities → what do you mean here?**
■ Different stem densities had been constructed in the study by Hu et al. (2014) and this research. This sentence has been modified:

In the study by Hu et al. (2014) and this research, cases are grouped by different densities

**31 P12.L238. Why are these considered as outliers?**
■ Experiments can have errors sometimes. I have revisited the original observations, the difference between Cd values from parallel experiments is relatively large for these two cases. Other symbols can also be outliers and they may affect the regression of data to some extent. This is unavoidable for experimental studies. We did not look into the outliers further, those two are just especially obvious. This sentence has been modified:

"These two trigons in the lower left corner were considered experimental outliers in these analyses."

**32 P12.L238. Tendencies → what do you mean with tendencies here?**
■ The original sentence has been edited:

"Results shows that for different groups of cases as the legend specified, there are obvious trend lines."

**33 P12.L240. Due to → be due to**
- It has been deleted.

**34 P12.L240. wave → waves**
- It has been deleted.

**35 P12.L240. This might…were small. → This sounds like a bit of guessing, but you should be able to check this.**
- This sentence has been deleted.

**36 P12.L240. Results revealed…was ignorable. → How do I see this? Which density differences?**
- Different stem densities had been constructed in the study by Hu et al. (2014) and this research. And the results had been separated in Fig. 7, see as VD1, VD2, VD3 for Hu et al. (2014) and N1, N2 for this research.

- Figure 7 have been edited (see the following figure) and the emergent condition of the cases can also be seen easily. Further, the calculated value of Cd by an equation from Wu et al. (2011) has been added in this figure to compare with our results.

- Additional Reference:
- He, F., Chen, J., Jiang C.: Surface wave attenuation by vegetation with the stem, root and canopy, Coast. Eng., 152, 103509, https://doi.org/10.1016/j.coastaleng.2019.103509, 2019.

[Figure]

**Figure 7: Relation between $R_e$ and the calculated $C_D$ by the new method. Different symbols indicate cases from different researches, and partially and fully solid symbols denote submerged and emerged cases, respectively. The solid lines following groups of the symbols indicate nonlinear fit by Eq. (13). The dashed dot line shows the calculated values by Eq. (15).**

■    The following paragraph has been added at the end of Section 2 and Fig. 8 has been modified:

Researchers had reported several formulas between $C_D$ and $Re$. For instance, Wu et al. (2011) obtained the following empirical equation:

$$C_d = 3.83 \times 10^{-6} + (5683/Re)^{1.17} \qquad (15)$$

Meanwhile, He et al. (2019) revealed that

$$C_d = 18.025 \exp(-0.043 KC) \qquad (16)$$

[Figure]

**Figure 8: Relation between $KC$ and the calculated $C_D$ by the new method. Details are the same as Fig. 7.**

**37 P12.L245. Various groups → do you mean the groups in Table 2? Please be specific.**
■    "varies groups" has been changed to "varies groups in Table 3".

**38 P12.L247. Case study is → case studies are?**
■    It has been edited.

**39 P15.L275. By reciprocal → by a reciprocal**
■    It has been edited.

**40 P15.L275. By combining...two perspectives. → These two reliable calibration methods by Dean (1979) and Kobayashi et al. (1993) can be combined from two perspectives:**
■    It has been edited.

**41 P15.L277. These relations → the relations**
- It has been edited.

**42 P15.L300. Filed → field?**
- Yes. It has been edited.

- Additional Reference:
- He, F., Chen, J., Jiang C.: Surface wave attenuation by vegetation with the stem, root and canopy, Coast. Eng., 152, 103509, https://doi.org/10.1016/j.coastaleng.2019.103509, 2019.

---

## Author Response (AR1)

We would like to thank referee #1 for the discussion.

The study of Zhang et al. focuses on wave attenuation by vegetation. Specifically, the authors look at an exponential and reciprocal function that describes wave height, and the accompanying damping factors. These damping factors both use the drag coefficient Cd, and the authors derive a new function that connects the two damping factors. Eventually, the authors predict Cd based on the different methods and damping factors. They conclude that the two damping factors are almost equal for submerged vegetation, but the new equation can be used for both submerged and emerged vegetation.

Generally, I like how the authors present their data, and find their conclusions well-supported with what they show. However, I still have several issues that the authors may want to improve on.

**1 Something that confuses me throughout the whole manuscript, is that several steps to get to the drag coefficient are not clearly described. For example, in the results section (sect. 5.1 and 5.3.3) the authors say they calculated alpha, but then the value of k should be known. So how was this done exactly? See also my minor comments for more examples. Generally, I think it would be good if the authors add one extra section in the methodology, where the method related to each section in the results is explained in more detail. There, the authors could state specifically, and maybe even step-wise, which equation was used in which way.**

- ■ Sentences (red marked) has been added or modified in the manuscript to show this methodology:

(1)P2,L41. Section 1. "These two calibration functions describe local wave height with a distance from the beginning of vegetation and a factor reflecting the damping, so the corresponding factor can be calibrated based on measured wave height through the vegetated area."

(2)P2,L45. Section 1. "Overall, the drag coefficient can be calculated by calibrating α' or k' using measured local wave height, then the researchers built non-liner relations between C_D and hydraulic parameters such as the Reynolds number (e.g., Hu et al., 2014; He et al., 2019). In this way, the drag of vegetation in water becomes predictable based on the non-linear relations and the values of these hydraulic parameters under different operating conditions."

(3)P7,L173. Section 5.1. "Additionally, with the calibrated k value from Eq. (8), we calculated the value of α according to Eq. (11). Applying the calculated α in Eq. (7), the calculated relative wave height, which was named by Eq. (11) in Fig. 2, was appliable to fit the measurements, which suggested that Eq. (11) is valid."

(4)P9,L199. Section 5.3.1. "Both methods by Dean (1979) and Dalrymple et al. (1984) consider wave height decaying by the reciprocal function, in which the damping factor can be obtained by fitting the local wave height by Eq. (7). In this case, the value of the drag coefficient can be calculated using Eq. (2) or Eq. (3), and the comparison of results by these two equations is

shown in Fig. 4."

(5)P10,L213. Section 5.3.2. "The value of C_D by Kobayashi et al. (1993) was obtained by calculating C_D using Eq. (5) on the base of the calibrated exponential damping factor by fitting the local wave height using Eq. (8). Figure 5 reveals that C_D by Eq. (5) is always smaller than C_D by Eq. (3)."

(6)P11,L223. Section 5.3.3. "The new method obtains the damping factor $\alpha'$ by using the calibrated $k'$ based on measured wave height and Eq. (12), so the drag coefficient $C_D$ can be calculated by Eq. (3)."

**2 I am also a bit confused about the Taylor expansions and how the authors arrive exactly at equations 11 and 12. This may be the lack of knowledge about this topic on my side, but I believe it is important to elaborate here and make it really clear to the reader what has been done and why.**

■ According the literature: the Taylor expansion is the standard technique used to obtain a linear or a quadratic approximation of a function of one variable. Recall that the Taylor expansion of a continuous function $f(x)$ is:

$$f(x) = f(a) + (x-a)f'(a) + (x-a)^2 \frac{f''(a)}{2!} + \cdots (x-a)^n \frac{f^{|n|}(a)}{n!} + \cdots R(x)$$

where $R(x)$ represents all the terms of higher order than a level, and a is a 'convenient' value at which to evaluate $f(x)$.

■ We scaled the reduction functions:

$$H/H_0 = 1/(1 + \alpha'X) = 1/(1 + \alpha x) = F(x), \tag{7}$$
$$H/H_0 = \exp(-k'X) = \exp(-kx) = G(x), \tag{8}$$

■ So by using Taylor expansion, Eqs. (7) and (8) can be derived as following when $x$ equals half:

$$F(x) = \frac{2}{\alpha+2} - \frac{4\alpha}{(\alpha+2)^2}(x-1/2) + \frac{8\alpha^2}{(\alpha+2)^3}(x-1/2)^2 - \frac{16\alpha^3}{(\alpha+2)^4}(x-1/2)^3 + R_1(x) \tag{9}$$

$$G(x) = \frac{1}{e^{k/2}} - \frac{k}{e^{k/2}}(x-1/2) + \frac{k^2}{2e^{k/2}}(x-1/2)^2 - \frac{k^3}{6e^{k/2}}(x-1/2)^3 + R_2(x), \tag{10}$$

■ Zhang et al. (2021) found that the first two terms of Eqs. (9) and (10) are relatively large compared to other terms. Hence, considering only these two terms in Eqs. (9) and (10), the proportionality between the two first terms:

$$\frac{2}{\alpha+2} / \frac{1}{e^{k/2}} = \frac{4\alpha}{(\alpha+2)^2}(x-1/2) / \frac{k}{e^{k/2}}(x-1/2)$$

■ Formula simplification results in:

$$\alpha/k = 2/(2-k), \tag{11}$$

which equals:

$$\alpha'/k' = 2/(2 - k'L), \tag{12}$$

Because $\alpha$ $(= \alpha'L)$ (-) is the scaled damping factor $\alpha'$, $k$ $(= k'L)$ (-) is the scaled exponential damping factor $k'$.

**3 I think the authors also need to elaborate the discussion. Especially the analyses to relate Cd to Re, KC and Ur are hardly discussed, and I think the authors should reflect here on the implications of their findings. Why is it important to link Cd to the parameters and what can we do with these findings? This is probably obvious for the authors, but it is good to also stress this for the reader.**

- ■ Sentences has been added or modified in the manuscript to show this answer this question:

- ■ (1)P2,L45. Section 1. "Overall, the drag coefficient can be calculated by calibrating α' or k' using measured local wave height, then the researchers built non-liner relations between C_D and hydraulic parameters such as the Reynolds number (e.g., Hu et al., 2014; He et al., 2019). In this way, the drag of vegetation in water becomes predictable based on the non-linear relations and the values of these hydraulic parameters under different operating conditions."

- ■ (2)The beginning of Section 5.4.1 in Page 12: "Relating the calculated $C_D$ by calibration method to $R_e$, $KC$, or $Ur$ is a common method to predict $C_D$."

**4 In addition, the authors should also make clear to the reader why the new equation is helpful and why the methods need to be linked. This is briefly done on page 16, but the main point seems here that you can use MS Excel, which is not a good argument in my view (there are so many free tools available, Python, R etc.). So please state clearly what new insights we gain from this and how it is helpful.**

- ■ The result based on Eq. (12) is comparable to the classic Eq. (3), this is because the effectiveness of Eq. (12) which we want to study further in this manuscript. Previously, Cd can be calibrated by Eq. (3) based on the reciprocal function, and by Eq. (5) and the exponential function. Now we found the latter method is only applicable in submerged cases. But we can also use the exponential function to describe the decay of the wave height for emerged cases and calculate the drag coefficient by transforming $k$ to alpha by Eq. (12).

- ■ The following paragraph has been added in Section 5.3.2, Page 10:

"For submerged cases, the drag coefficient by Eq. (5) is close to but slightly smaller than that by Eq. (3), with a slope of 0.96 in Fig. 5; for emerged cased, the former is more smaller than the latter when the drag coefficient is larger. This is consistent to the conclusion in Section 5.2 since $C_D$ has positive correlation with $\alpha$ and $k$."

■ The following paragraph has been added at the end of Section 5.3.3, Page 11:

"Based on the results in Figs. 5 and 6, the exponential damping factor $k'$ can be used to calculate $C_D$ while it needs to be converted to $\alpha'$ based on Eq. (12) instead of to be used directly in Eq. (5) for emerged cases; while for submerged cases, it can be a solution to calculate $C_D$ directly.."

■ Please see the reply to Reviewer#2 for more detail.

**5 Lastly, my list of minor issues is still rather long and probably not even complete. Therefore, I would suggest that the authors go over the article again in full detail and try to improve their text.**

■ Thanks for the correction of language. The language has been checked and will be checked further.

To conclude, I like the study and believe the results are clear. Most of my comments are merely textual, so I believe the authors could easily improve the manuscript. I hope the authors find my comments useful, and I am looking forward to a revised version of the manuscript.

**Minor comments**
Abstract → The abstract should not refer to the main text, so it is better to remove the equation
numbers.
**1 P1.L16. Predicting → predict**

■ It has been modified to "calculate".

**2 P1.L24 of practical → of practical use?**

■ It has been modified to "it is practical to …".

**3 P1.L24 barrier→ barriers**

■ It has been edited.

**4 P1.L29. What do you mean with floodplain resources?**

■ It has been modified to "land resources in floodplain".

**5 P2.L32. Water motion in researches → water motion, as investigated in different researches**

■ It has been edited.

**6 P2.L36. Complicate → complicated    P2.L39. Following → follow a**
- ■ It has been edited.

**7 P2.L44. Later → that?**
- ■ It has been deleted.

**8 P2.L47. then→ that?**
- ■ No modification required.

**9 P3.L64. Vertical, rigid cylinder → a vertical, rigid cylinder / vertical, rigid cylinders ?**
- ■ It has been modified to "vertical, rigid cylinders".

**10 P3.L66. Of circular → of the circular**
- ■ It has been edited.

**11 P4.L100. What do you mean with the proportionality? I am not sure if I follow how you get to Eq. 11.**
- ■ This sentence has been edited to:

"the proportionality $\frac{2}{\alpha+2}/\frac{1}{e^{k/2}} = \frac{4\alpha}{(\alpha+2)^2}(x-1/2)/\frac{k}{e^{k/2}}(x-1/2)$  results in:"

**12 P4.L112. Understanding → understand**
- ■ It has been edited.

**13 P4.L120. Why do you use that specific formula?**
- ■ The last part of Section 2 has been edited:

"Researchers had reported several formulas between $C_D$ and $Re$. For instance, Wu et al. (2011) obtained the following empirical equation:

$$C_d = 3.83 \times 10^{-6} + (5683/Re)^{1.17} \tag{13}$$

Besides, He et al. (2019) revealed that

$$C_d = 18.025 \exp(-0.043KC) \tag{14}$$

Hence, the following two formulas are most possible solutions to study the non-linear relation between $C_D$ and these parameters:

$$C_D = a \exp(-b\bar{X}) \tag{15}$$
$$C_D = a + (b/\bar{X})^c \tag{16}$$

where $\bar{X}$ could be $R_e$, $KC$ or $Ur$; $a$, b, c are factors. Values of these factors can be obtained by the regression of $C_D$ by calibrated $\alpha'$ or $k'$ and these parameters, and in this way, $C_D$ becomes predictable under different operation conditions. We had obtained the values of the factors and the corresponding adjusted R-square as in Section 5.4 by both equations, and it is hard to tell the difference between these results from Eqs. (13) and (14). The former is at last chosen because it contains less factors and is simpler than the latter."

**14 P4.L131. The three lengths of the canopies → here you mean the horizontal length, correct? So, x in Figure 1?**

■ Yes. "The three lengths of the canopies" has been modified to "The three horizontal lengths of the vegetated area".

**15 P4.L132. these → the**

■ It has been edited.

**16 P4.L132. Depth → depths**

■ It has been edited.

**17 P4.L140. List → listed**

■ It has been edited.

**18 P5.L141. Collected more → collected during more?**

■ Yes. It has been edited.

**19 P7.L151. Had shown → determined?**

■ It has been deleted and this paragraph has been modified to:

"Besides experiments in this study, observations in published literatures had been collected from Hu et al. (2014), Wu et al. (2011), and Wu and Cox (2015, 2016) as Zhang et al. (2021) presented. The summarized experimental setup is shown in Table 2. Overall, different operation conditions had been conducted by the researches."

**20 P7.L158. In laboratory → in a laboratory**

■ It has been deleted.

**21 P7.L173-174. Also...useful.→ How did you use Equation 11 here? How did you determine the k-value?**

■ It has been edited to.

"Additionally, with the calibrated k value from Eq. (8), we calculated the value of α according to Eq. (11). Applying the calculated α in Eq. (7), the calculated relative wave height, which was named by Eq. (11) in Fig. 2, was appliable to fit the measurements, which suggested that Eq. (11) is valid."

**22 P8.L180. Was shown → is shown**

■ It has been edited.

**23 P8.L187. did not strongly affect → was not strongly affected**

■ It has been modified to "is not strongly affected".

**24 P9.L198. Attention..recently.-→ Several studies paid attention to the emergent condition of the vegetation recently.**

■ It has been edited.

**25 P9.L199. In this part...were compared. → How do you do this exactly? You fit equation 1 for alpha, and then calculate the drag coefficient Cd back with Eqs 2 and 3?**

■ Yes. The typical process to get the drag coefficient by the calibration method is like this: measure the local wave height H(X) through the vegetated area then calibrate the value of alpha or $k$ by Eqs. (7) or (8). Then Cd can be calculated by equations such as Eqs. (2), (3), (5). This is the calibrated value of Cd. Then the authors often relate the calibrated Cd to Re or KC, so the Cd is predictable when lack of measurement data. It has been modified: "Both methods by Dean (1979) and Dalrymple et al. (1984) consider wave height decaying by the reciprocal function, in which the damping factor can be obtained by fitting the local wave height by Eq. (7). In this case, the value of the drag coefficient can be calculated using Eq. (2) or Eq. (3), and the comparison of results by these two equations is shown in Fig. 4."

**26 P9.L204. When study → when studying**

■ It has been deleted.

**27 P9.L213. Decaying function → decaying functions**

■ It has been edited.

**28 P11.L223. How did you determine k here?**

■ k is obtained by calibrating the measured wave height through the vegetation by Eq. (8). This sentence has been modified: "The new method obtains the damping factor $\alpha'$ by using the calibrated $k'$ based on measured wave height and Eq. (12), so the drag coefficient $C_D$ can be calculated by Eq. (3)."

**29 P11.L224. Decaying → decays**

■ It has been edited.

**30 P12.L237. Different densities → what do you mean here?**

■ Different stem densities had been constructed in the study by Hu et al. (2014) and this research. This sentence has been modified: "In the study by Hu et al. (2014) and this research, cases were grouped by different densities."

**31 P12.L238. Why are these considered as outliers?**

■ Experiments can have errors sometimes. I have revisited the original observations, the difference between Cd values from parallel experiments is relatively large for these two cases. Other symbols can also be outliers and they may affect the regression of data to some extent. This is unavoidable for experimental studies. We did not look into the outliers further, those two are just especially obvious. However, this sentence has been deleted, and the result in Tables 3 and 4 have been edited.

■ Also, the last sentence of the manuscript has been modified to: "however, the interaction between the vegetation and flow field is complicated and laboratory errors may affect the result so verification and/or calibration are needed further for predicting the drag coefficient."

**32 P12.L238. Tendencies → what do you mean with tendencies here?**

■ The original sentence has been deleted and the following sentence has been modified: "The values of $R_e$ ranged from 370 to 38000 and the solid line following different groups of symbols can basically fit,"

**33 P12.L240. Due to → be due to**

■ It has been deleted.

**34 P12.L240. wave → waves**

■ It has been deleted.

**35 P12.L240. This might...were small. → This sounds like a bit of guessing, but you should be able to check this.**

■ This sentence has been deleted.

**36 P12.L240. Results revealed...was ignorable. → How do I see this? Which density differences?**

■ This sentence has been modified to: "Results reveals that separating cases from different densities is necessary for studying this relation while the effect of the emergent condition can be ignorable."

■ Different stem densities had been constructed in the study by Hu et al. (2014) and this research. And the results had been separated in Fig. 7, see as VD1, VD2, VD3 for Hu et al. (2014) and N1, N2 for this research.

■ Figures 7, 8, 9 have been edited (see the following figures) and the emergent condition of the cases can also be seen easily. Further, the calculated value of $C_D$ by an equation from Wu et al. (2011) has been added in this figure to compare with our results.

[Figure]

**Figure 7: Relation between $R_e$ and $C_D$ by the new method. Different symbols indicate cases from different researches, and partially and fully solid symbols denote submerged and emerged cases, respectively. The solid lines following groups of the symbols indicate nonlinear fit by Eq. (15).**

[Figure]

**Figure 8: Relation between $KC$ and the calculated $C_D$ by the new method. Details are the same as Fig. 7.**

[Figure]

**Figure 9: Relation between $Ur$ and the calculated $C_D$ by the new method. Details are the same as Fig. 7.**

**37 P12.L245. Various groups → do you mean the groups in Table 2? Please be specific.**
■ "varies groups" has been changed to "varies groups in Table 3".

**38 P12.L247. Case study is → case studies are?**
■ It has been edited.

**39 P15.L275. By reciprocal → by a reciprocal**
■ It has been edited.

**40 P15.L275. By combining...two perspectives. → These two reliable calibration methods by Dean (1979) and Kobayashi et al. (1993) can be combined from two perspectives:**
■ It has been edited.

**41 P15.L277. These relations → the relations**
■ It has been edited.

**42 P15.L300. Filed → field?**
■ Yes. It has been edited.

We would like to thank referee #2 for the discussion.

In this study, the authors proposed a new hybrid method to link the two damping factors derived from two traditional approaches. Subsequently, the method was used to calibrate the drag coefficient and the relationship between the drag coefficient and relevant parameters (Re, KC, and Ur) were investigated. The paper is generally well-written. However, there are several major concerns that should be properly addressed before the paper can be considered to be accepted by this journal.

The major concerns:

1. The novelty of the manuscript: the authors mentioned that "Besides, based on local wave height, the exponential damping factor $k'$ can be obtained easily by MS Excel, while the damping factor $\alpha'$ needs professional numerical tools. Therefore, calculating $\alpha'$ by the calibrated $k'$ is much easier than calibrating $\alpha'$ directly by the well documented Eq. (3) which is the advantage of the new method in this study." I agree with the comments provided by the Reviewer#1 that this should not be the main novelty of this manuscript since the calibration of the damping factor $\alpha'$ is a standard procedure and can be easily conducted by commonly used software (such as Matlab or R language).

- ■ This paragraph has been deleted. The novelty of the manuscript can be seen in the reply for the second question.

2. The methodology: It appears that the key Equations (7)-(12) in this manuscript have been derived in the previous study by the authors (Zhang et al., Acta Oceanol. Sin, 2021, in press). Thus, the main contribution lies in the study of the relation between the drag coefficient and three relevant hydraulic parameters? I would suggest the authors to clarify the relationship between their previous study and the current paper.

- ■ (1)Our previous study overlooked the relation between $k'$ and $C_D$ by Kobayashi et al. (1993), and the current study uses this relation. (2)The previous study only used the relation between $\alpha'$ and $C_D$ by Dean (1979) and the relation between $\alpha'$ and $C_D$ by Dalrymple et al. (1984) has been used in this study.

- ■ About the equations, there are several improvements: (1)we found Eq. (12) can be the relation between $\alpha'$ and $k'$ so in this study we tried to prove it further such as in Section 5.1, 5.2, 5.3.3; (2)more data has been collected and we constructed experiments by ourselves; (3)the emergent condition is considered to study the applicability of Eq. (12); (4)the equation has been analyzed further.

- ■ P2,L45. The paragraph has been modified:

"Zhang et al. (2021) had compared these two calibration approaches by these two featured functions directly and yielded a connection between $\alpha'$ and $k'$, then a new equation to calculate the drag coefficient had been revealed. However, Zhang et al. (2021) overlooked the relation between $k'$ and $C_D$ by Kobayashi et al. (1993) and only used the relation between $\alpha'$ and $C_D$ by Dean (1979). In this article, using the well documented relation between the

damping factor $\alpha'$ and the drag coefficient $C_D$ by Dalrymple et al. (1984) as well as the mentioned relation by Kobayashi et al. (1993), these two traditional approaches had been compared from another perspective and the second connection between $\alpha'$ and $k'$ had been revealed.

Hence, there are two relations between the damping factor and the exponential damping factor from two perspectives, and they had been analyzed by 99 cases from collected data and experimental experiments in this study."

■ The following paragraph has been added in the end of Section 5.2 in Page 8 to analyze the equation with data:

"Equation (12) also revealed that $\alpha - k = k^2/(2-k) > 0$ since $k$ is smaller than 2 (Fig. 3). When the vegetation is deeply submerged, the calibrated $k$ close to zero and $\alpha$ is larger than but approximate to $k$ (Eq. (6)); when the vegetation becomes emerged, $\alpha$ and $k$ become relatively large and the difference between them enlarges, which can be seen in Figs. 2 and 3. That is to say, Fig. 3 shows that Eq. (12) works well and it includes Eq. (6) to some extent."

■ The following paragraph has been added in the end of Section 5.3.2 in Page 10 to analyze the equation with data:

"Additionally, although the regression of data should not be linear since $k/\alpha = (2-k)/2 < 1$ is not a constant, if we obtain $C_D$ by calibrating the exponential function for emerged cases, we have a rapid assessment that the value will be approximate 77% of the needed value. Moreover, the result reveals that $k'/\alpha' \approx 0.77$. Combining Eq. (12), $k'L = k$ approximates to 0.46, then $K_X \approx 0.63$ at the end of the vegetation according to Eqs. (4) and (8). It means that the reduction rate ($=1-K_X$) of the wave height for the emerged cases is about 37%. Furthermore, if we apply $k \approx 0.46$ in Eq. (12), $\alpha$ is about 0.53 then $K_X \approx 0.65$ according to Eqs. (1) and (7). Values of $K_X$ which were close by $\alpha$ and $k$ can be used to assess the wave attenuation by emerged vegetation very preliminary.

Of course, several parameters can affect the drag effect. In this case, certain cases should be considered instead of to use the result from a regression by all the cases with different operating conditions, then the slope of the comparison between the calculated $C_D$ by Eqs. (3) and (5) will be different so the calculated relative wave height will be different."

■ Besides, we looked at the relation between Cd and the hydraulic parameter and revealed that it is not easy to find a simple formula to describe the relation based on data of 99 cases. We also compared different methods for calculating Cd and found the limitation of the methods by Dean (1979) and Kobayashi et al. (1993).

3. Figure 6: It appears that the proposed new method (Eq. 12) functions more or less the same as Eq. (3). Thus, with regard to the calibration of the drag coefficient, what's the difference between the new method and the method proposed by Dalrymple et al. (1984)?

- The result based on Eq. (12) is comparable to the classic Eq. (3), this is because the effectiveness of Eq. (12) which we want to study further in this manuscript. Previously, Cd can be calibrated by Eq. (3) based on the reciprocal function, and by Eq. (5) and the exponential function. Now we found the latter method is only applicable in submerged cases. But we can also use the exponential function to describe the decay of the wave height for emerged cases and calculate the drag coefficient by transforming $k$ to alpha by Eq. (12).

- The following paragraph has been added in Section 5.3.2, Page 10:

"For submerged cases, the drag coefficient by Eq. (5) is close to but slightly smaller than that by Eq. (3), with a slope of 0.96 in Fig. 5; for emerged cased, the former is more smaller than the latter when the drag coefficient is larger. This is consistent to the conclusion in Section 5.2 since $C_D$ has positive correlation with $\alpha$ and $k$."

- The following paragraph has been added at the end of Section 5.3.3, Page 11:

"Based on the results in Figs. 5 and 6, the exponential damping factor $k'$ can be used to calculate $C_D$ while it needs to be converted to $\alpha'$ based on Eq. (12) instead of to be used directly in Eq. (5) for emerged cases; while for submerged cases, it can be a solution to calculate $C_D$ directly."

4. The underlying mechanism and the difference between emerged and submerged conditions: one possible novelty could be the unified expression for the calibration of the drag coefficient both emerged and submerged conditions. However, can authors further explore the underlying mechanism and the difference between these two conditions by means of the new proposed method?

- Equation (3) is a well performed equation to calibrate Cd directly. One the other hand, the following paragraph at the end of Section 5.2 reveals Eq. (5) can only be valid for submerged conditions.:

"Notably, the analytical solution of Kobayashi et al. (1993), i.e., Eq. (5), was obtained and conducted using deeply submerged artificial kelp, and $H(X)^3 \cong H_0 H(X)^2$ was assumed which can only be valid when wave height reduces slightly through submerged vegetated areas and the exponential damping factor is small. This is why Eq. (6) can only be profitable for submerged vegetation."

- If we compare Eqs. (3) and (5) directly and get Eq. (6) which can also be useful when the vegetation is submerged. In other words ,the combination of Eq. (5) and the exponential function has limitation to get the value of Cd. However, the exponential function can also describe the decay of the wave height, if we want to get Cd by the exponential decay of wave height, the relation of Eq. (12) is needed for emerged cases.

■ We will try to study the underlying mechanism and the difference between emerged and submerged conditions from mathematic approach to look at Eqs. (7) and (8) and use the relation between alpha and k.

The minor comments:

1. Please carefully address all the minor comments provided by Reviewer#1.

■ It has been edited.

2. Abstract: both equations and symbols should be avoided.

■ The abstract has been modified:

"Vegetation in wetlands is a large-scale nature-based resource providing a myriad of services for human beings and the environment, such as dissipating incoming wave energy and protecting coastal areas. For understanding wave height attenuation by vegetation, there are two main traditional calibration approaches to the drag effect acting on the vegetation. One of them is based on the rule that wave height decays through the vegetated area by a reciprocal function and another by an exponential function. In both functions, the local wave height reduces with distance from the beginning of the vegetation depending on a damping factor. These two damping factors which are usually obtained from calibration by measured local wave height are linked to the drag coefficient and measurable parameters, respectively. So the drag coefficient that quantifies the effect of the vegetation can be calculated by different methods, following by connecting this coefficient to hydraulic parameters to make it predictable. In this study, two relations between these two damping factors and methods to calculate the drag coefficient had been investigated by 99 laboratory experiments. Finally, relations between the drag coefficient and relevant parameters were analyzed. The results show that emergent conditions should be considered when studying the drag coefficient; traditional methods which had overlooked this condition cannot perform well when the vegetation was emerged. The new method based on the relation between these two damping factors performed as well as the well-recognized method for emerged and submerged vegetation. Additionally, the Keulegan-Carpenter number can be a suitable hydraulic parameter to predict the drag coefficient only the experimental setup especially the densities of the vegetation can affect the prediction equations."

3. Figures 4-9: in both xlabel and ylabel, the Cd should be corrected as CD

■ It has been edited.

4. Section 4 data collection: Please reorganize this section, for the time being, the authors simply
list the collected data.

■ Section 4 has been modified:

"Besides experiments in this study, observations in published literatures had been collected from Hu et al. (2014), Wu et al. (2011), and Wu and Cox (2015, 2016) as Zhang et al. (2021)

presented. The summarized experimental setup is shown in Table 2. Overall, different operation conditions had been conducted by the researches.

**Table 2: Experimental conditions from references**

| Reference | Type of plant | Plant height/m | Plant diameter/m | Plant density/ stem m⁻² | Incident wave height/m | Length of vegetation/m | Depth of water/m |
|---|---|---|---|---|---|---|---|
| Hu et al. (2014) | Stiff wooden rods | 0.36 | 0.01 | 62/139/556 (VD1/VD2/VD3) | 0.032~0.202 | 6 | 0.25/0.5 |
| Wu et al. (2011) | Birch dowels | 0.48/0.63 | 0.009 4 | 350/623 | 0.083/0.084/0.085 | 3.66 | 0.5 |
| Wu and Cox (2015) | Plastic strips | 0.14 | 0.005 | 2 100 | 0.014~0.042 | 1.8 | 0.12 |
| Wu and Cox (2016) | Plastic strips | 0.14 | 0.005 | 1 618 | 0.015~0.034 | 0.9 | 0.12 |

5. Figure 3: in the legend, "Calculted"
- ■ It has been edited.

- ■ Additional Reference:
- ■ He, F., Chen, J., Jiang C.: Surface wave attenuation by vegetation with the stem, root and canopy, Coast. Eng., 152, 103509, https://doi.org/10.1016/j.coastaleng.2019.103509, 2019.

---

## Author Response (AR2)

We would like to thank the handling editor for the decision.

There are several correlations we have made:

1) We have redrawn Figs. 3-7, mainly by separating the submerged cases from emerged cases to make them readable.

2) Numbers of equations have been checked and edited thoroughly.

3) Results in Section 5.3.2 have been checked and edited (Page 12, Line 249).

4) Affiliations and fundings have been checked and edited (Page1, Line6 and Page 18, Line 352). The sections "Data availability" and "Competing interests" have been included (Page 18).

5) Minor corrections about the language.